# Learning Cut Selection for Mixed-Integer Linear Programming via Hierarchical Sequence Model

**Zhihai Wang**[*1] , **Xijun Li**[*1,2], **Jie Wang**[†1,3] , **Yufei Kuang**[1], **Mingxuan Yuan**[2],
**Jia Zeng**[2], **Yongdong Zhang**[1,3], **Feng Wu**[1,3]
[1] CAS Key Laboratory of Technology in GIPAS, University of Science and Technology of China
[2] Noah's Ark Lab, Huawei Technologies
[3] Institute of Artificial Intelligence, Hefei Comprehensive National Science Center

## Abstract

Cutting planes (cuts) are important for solving mixed-integer linear programs (MILPs), which formulate a wide range of important real-world applications. Cut selection—which aims to select a proper subset of the candidate cuts to improve the efficiency of solving MILPs—heavily depends on **(P1)** which cuts should be preferred, and **(P2)** how many cuts should be selected. Although many modern MILP solvers tackle **(P1)-(P2)** by manually designed heuristics, machine learning offers a promising approach to learn more effective heuristics from MILPs collected from specific applications. However, many existing learning-based methods focus on learning which cuts should be preferred, neglecting the importance of learning the number of cuts that should be selected. Moreover, we observe from extensive empirical results that **(P3)** what *order of selected cuts* should be preferred has a significant impact on the efficiency of solving MILPs as well. To address this challenge, we propose a novel **h**ierarchical s**e**quence **m**odel (HEM) to learn cut selection policies via reinforcement learning. Specifically, HEM consists of a two-level model: (1) a higher-level model to learn the number of cuts that should be selected, (2) and a lower-level model—that formulates the cut selection task as a sequence to sequence learning problem—to learn policies selecting an *ordered subset* with the size determined by the higher-level model. To the best of our knowledge, HEM is *the first* method that can tackle **(P1)-(P3)** in cut selection simultaneously from a data-driven perspective. Experiments show that HEM significantly improves the efficiency of solving MILPs compared to human-designed and learning-based baselines on both synthetic and large-scale real-world MILPs, including MIPLIB 2017. Moreover, experiments demonstrate that HEM well generalizes to MILPs that are significantly larger than those seen during training. Code is available at `https://github.com/MIRALab-USTC/L2O-HEM-Torch` (PyTorch version), and `https://gitee.com/mindspore/models/tree/master/research/l2o/hem-learning-to-cut` (MindSpore version).

## 1 Introduction

Mixed-integer linear programming (MILP) is a general optimization formulation for a wide range of important real-world applications, such as supply chain management (Paschos, 2014), production planning (Jünger et al., 2009), scheduling (Chen, 2010), facility location (Farahani & Hekmatfar, 2009), bin packing (Nair et al., 2020), etc. A standard MILP takes the form of

$$z^* \triangleq \min_{\mathbf{x}} \{\mathbf{c}^\top \mathbf{x} | \mathbf{A}\mathbf{x} \leq \mathbf{b}, \mathbf{x} \in \mathbb{R}^n, x_j \in \mathbb{Z} \text{ for all } j \in \mathcal{I}\}, \quad (1)$$

where $\mathbf{c} \in \mathbb{R}^n$, $\mathbf{A} \in \mathbb{R}^{m \times n}$, $\mathbf{b} \in \mathbb{R}^m$, $x_j$ denotes the $j$-th entry of vector $\mathbf{x}$, $\mathcal{I} \subseteq \{1, \dots, n\}$ denotes the set of indices of integer variables, and $z^*$ denotes the optimal objective value of the problem

---

*Equal contribution. This work was done when Zhihai Wang was an intern at Huawei Noah's Ark Lab.
†Corresponding author: jiewangx@ustc.edu.cn

in (1). However, MILPs can be extremely hard to solve as they are $\mathcal{NP}$-hard problems (Bixby et al., 2004). To solve MILPs, many modern MILP solvers (Gurobi, 2021; Bestuzheva et al., 2021; FICO Xpress, 2020) employ a branch-and-bound tree search algorithm (Land & Doig, 2010), in which a linear programming (LP) relaxation of a MILP (the problem in (1) or its subproblems) is solved at each node. To further enhance the performance of the tree search algorithm, cutting planes (cuts) (Gomory, 1960) are introduced to tighten the LP relaxations (Achterberg, 2007; Bengio et al., 2021). Existing work on cuts falls into two categories: cut generation and cut selection (Turner et al., 2022). Cut generation aims to generate cuts, i.e., valid linear inequalities that tighten the LP relaxations (Achterberg, 2007). However, adding all the generated cuts to the LP relaxations can pose a computational problem (Wesselmann & Stuhl, 2012). To further improve the efficiency of solving MILPs, cut selection is proposed to select a proper subset of the generated cuts (Wesselmann & Stuhl, 2012). In this paper, we focus on **the cut selection problem**, which has a significant impact on the overall solver performance (Achterberg, 2007; Tang et al., 2020; Paulus et al., 2022).

Cut selection heavily depends on **(P1)** which cuts should be preferred, and **(P2)** how many cuts should be selected (Achterberg, 2007; Dey & Molinaro, 2018b). Many modern MILP solvers (Gurobi, 2021; Bestuzheva et al., 2021; FICO Xpress, 2020) tackle **(P1)-(P2)** by hard-coded heuristics designed by experts. However, hard-coded heuristics do not take into account underlying patterns among MILPs collected from certain types of real-world applications, e.g., day-to-day production planning, bin packing, and vehicle routing problems (Pochet & Wolsey, 2006; Laporte, 2009; Nair et al., 2020). To further improve the efficiency of MILP solvers, recent methods (Tang et al., 2020; Paulus et al., 2022; Huang et al., 2022) propose to learn cut selection policies via machine learning, especially reinforcement learning. They offer promising approaches to learn more effective heuristics by capturing underlying patterns among MILPs from specific applications (Bengio et al., 2021). However, many existing learning-based methods (Tang et al., 2020; Paulus et al., 2022; Huang et al., 2022)—which learn a scoring function to measure cut quality and select a fixed ratio/number of cuts with high scores—suffer from two limitations. First, they learn which cuts should be preferred by learning a scoring function, neglecting the importance of learning the number of cuts that should be selected (Dey & Molinaro, 2018b). Moreover, we observe from extensive empirical results that **(P3)** what *order of selected cuts* should be preferred significantly impacts the efficiency of solving MILPs as well (see Section 3). Second, they do not take into account the interaction among cuts when learning which cuts should be preferred, as they score each cut *independently*. As a result, they struggle to select cuts that complement each other nicely, which could severely hinder the efficiency of solving MILPs (Dey & Molinaro, 2018b). Indeed, we empirically show that they tend to select many similar cuts with high scores (see Experiment 4 in Section 5).

To address the aforementioned challenges, we propose a novel **h**ierarchical s**e**quence **m**odel (HEM) to learn cut selection policies via reinforcement learning. To the best of our knowledge, HEM is *the first* learning-based method that can tackle **(P1)-(P3)** simultaneously by proposing a two-level model. Specifically, HEM is comprised of (1) a higher-level model to learn the number of cuts that should be selected, (2) and a lower-level model to learn policies selecting an *ordered subset* with the size determined by the higher-level model. The lower-level model formulates the cut selection task as a sequence to sequence learning problem, leading to two major advantages. First, the sequence model is popular in capturing the underlying order information (Vinyals et al., 2016), which is critical for tackling **(P3)**. Second, the sequence model can well capture the *interaction* among cuts, as it models the *joint* conditional probability of the selected cuts given an input sequence of the candidate cuts. As a result, experiments show that HEM significantly outperforms human-designed and learning-based baselines in terms of solving efficiency on three synthetic MILP problems and seven challenging MILP problems. The challenging MILP problems include some benchmarks from MIPLIB 2017 (Gleixner et al., 2021) and large-scale real-world production planning problems. Our results demonstrate the strong ability to enhance modern MILP solvers with our proposed HEM in real-world applications. Moreover, experiments demonstrate that HEM can well generalize to MILPs that are significantly larger than those seen during training.

## 2 BACKGROUND

**Cutting planes.** Given the MILP problem in (1), we drop all its integer constraints to obtain its *linear programming (LP) relaxation*, which takes the form of

$$z^*_{\text{LP}} \triangleq \min_{\mathbf{x}} \{\mathbf{c}^\top \mathbf{x} | \mathbf{A}\mathbf{x} \leq \mathbf{b}, \mathbf{x} \in \mathbb{R}^n\}. \tag{2}$$

Since the problem in (2) expands the feasible set of the problem in (1), we have $z^*_{\text{LP}} \leq z^*$. We denote any lower bound found via an LP relaxation by a *dual bound*. Given the LP relaxation in (2), cutting planes (cuts) are linear inequalities that are added to the LP relaxation in the attempt to tighten it without removing any integer feasible solutions of the problem in (1). Cuts generated by MILP solvers are added in successive rounds. Specifically, each round $k$ involves (i) solving the current LP relaxation, (ii) generating a pool of candidate cuts $\mathcal{C}^k$, (iii) selecting a subset $\mathcal{S}^k \subseteq \mathcal{C}^k$, (iv) adding $\mathcal{S}^k$ to the current LP relaxation to obtain the next LP relaxation, (v) and proceeding to the next round. Adding all the generated cuts to the LP relaxation would maximally strengthen the LP relaxation and improve the lower bound at each round. However, adding too many cuts could lead to large models, which can increase the computational burden and present numerical instabilities (Wesselmann & Stuhl, 2012). Therefore, cut selection is proposed to select a proper subset of the candidate cuts, which is significant for improving the efficiency of solving MILPs (Tang et al., 2020).

**Branch-and-cut.** In modern MILP solvers, cutting planes are often combined with the branch-and-bound algorithm (Land & Doig, 2010), which is known as the branch-and-cut algorithm (Mitchell, 2002). Branch-and-bound techniques perform implicit enumeration by building a search tree, in which every node represents a subproblem of the original problem in (1). The solving process begins by selecting a leaf node of the tree and solving its LP relaxation. Let $\mathbf{x}^*$ be the optimal solution of the LP relaxation. If $\mathbf{x}^*$ violates the original integrality constraints, two subproblems (child nodes) of the leaf node are created by *branching*. Specifically, the leaf node is added with constraints $x_i \leq \lfloor x^*_i \rfloor$ and $x_i \geq \lceil x^*_i \rceil$, respectively, where $x_i$ denotes the $i$-th variable, $x^*_i$ denotes the $i$-th entry of vector $\mathbf{x}^*$, and $\lfloor \rfloor$ and $\lceil \rceil$ denote the floor and ceil functions. In contrast, if $\mathbf{x}^*$ is a (mixed-)integer solution of (1), then we obtain an upper bound on the optimal objective value of (1), which we denote by *primal bound*. In modern MILP solvers, the addition of cutting planes is alternated with the *branching* phase. That is, cuts are added at search tree nodes before branching to tighten their LP relaxations. Since strengthening the relaxation before starting to branch is decisive to ensure an efficient tree search (Wesselmann & Stuhl, 2012; Bengio et al., 2021), we focus on only adding cuts at the root node, which follows Gasse et al. (2019); Paulus et al. (2022).

**Primal-dual gap integral.** We keep track of two important bounds when running branch-and-cut, i.e., the global primal and dual bounds, which are the best upper and lower bounds on the optimal objective value of (1), respectively. We define the *primal-dual gap integral* (PD integral) by the *area* between the curve of the solver's global primal bound and the curve of the solver's global dual bound. We provide more details in Appendix C.1.

## 3 MOTIVATING RESULTS

We empirically show that the *order of selected cuts*, i.e., the selected cuts are added to the LP relaxations in this order, significantly impacts the efficiency of solving MILPs. Moreover, we empirically show that the *ratio of selected cuts* matters significantly when solving MILPs (see Appendix G.1). Please see Appendix D.2 for details of the datasets used in this section.

**Order matters.** Previous work (Bixby, 1992; Maros, 2002; Li et al., 2022) has shown that the order of constraints for a given linear program (LP) significantly

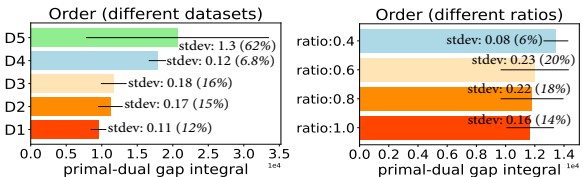

(a) Evaluate RandomAll on five different datasets.

(b) Evaluate RandomNv on MIPLIB mixed neos.

Figure 1: We design two cut selection heuristics, namely RandomAll and RandomNV (see Section 3 for details), which both add the same subset of cuts in random order for a given MILP. The results in (a) and (b) show that adding the same selected cuts in different order leads to variable overall solver performance.

impacts its constructed initial basis, which is important for solving the LP. As a cut is a linear constraint, adding cuts to the LP relaxations is equivalent to adding constraints to the LP relaxations. Therefore, the order of added cuts could have a significant impact on solving the LP relaxations as well, thus being important for solving MILPs. Indeed, our empirical results show that this is the case. (1) We design a **RandomAll** cut selection rule, which randomly permutes all the candidate cuts, and adds all the cuts to the LP relaxations in the random order. We evaluate RandomAll on five challenging datasets, namely D1, D2, D3, D4, and D5. We use the SCIP 8.0.0 (Bestuzheva et al., 2021) as the backend solver, and evaluate the solver performance by the average PD integral within a time limit. We evaluate RandomAll on each dataset over ten random seeds, and each bar in Figure

1a shows the mean and standard deviation (stdev) of its performance on each dataset. As shown in Figure 1a, the performance of RandomAll on each dataset varies widely with the order of selected cuts. (2) We further design a **RandomNV** cut selection rule. RandomNV is different from RandomAll in that it selects a given ratio of the candidate cuts rather than all the cuts. RandomNV first scores each cut using the Normalized Violation (Huang et al., 2022) and selects a given ratio of cuts with high scores. It then randomly permutes the selected cuts. Each bar in Figure 1b shows the mean and stdev of the performance of RandomNV with a given ratio on the same dataset. Figures 1a and 1b show that adding the same selected cuts in different order leads to variable solver performance, which demonstrates that the order of selected cuts is important for solving MILPs.

## 4 LEARNING CUT SELECTION VIA HIERARCHICAL SEQUENCE MODEL

In the cut selection task, the optimal subsets that should be selected are inaccessible, but one can assess the quality of selected subsets using a solver and provide the feedbacks to learning algorithms. Therefore, we leverage reinforcement learning (RL) to learn cut selection policies. In this section, we provide a detailed description of our proposed RL framework for learning cut selection. First, we present our formulation of the cut selection as a Markov decision process (MDP) (Sutton & Barto, 2018). Then, we present a detailed description of our proposed **h**ierarchical s**e**quence **m**odel (HEM). Finally, we derive a hierarchical policy gradient for training HEM efficiently.

**Reinforcement Learning Formulation**

As shown in Figure 2, we formulate a MILP solver as the environment and our proposed HEM as the agent. We consider an MDP defined by the tuple $(\mathcal{S}, \mathcal{A}, r, f)$. Specifically, we specify the state space $\mathcal{S}$, the action space $\mathcal{A}$, the reward function $r : \mathcal{S} \times \mathcal{A} \rightarrow \mathbb{R}$, the transition function $f$, and the terminal state in the following. **(1) The state space $\mathcal{S}$.** Since the current LP relaxation and the generated cuts contain the core information for cut selection, we define a state $s$ by $(M_{\text{LP}}, \mathcal{C}, \mathbf{x}_{\text{LP}}^*)$. Here $M_{\text{LP}}$ denotes the mathematical model of the current LP relaxation, $\mathcal{C}$ denotes the set of the candidate cuts, and $\mathbf{x}_{\text{LP}}^*$ denotes the optimal solution of the LP relaxation. To encode the state information, we follow Achterberg (2007); Huang et al. (2022) to design thirteen features for each candidate cut

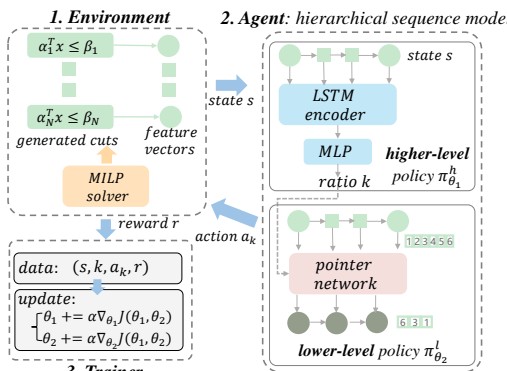

Figure 2: Illustration of our proposed RL framework for learning cut selection policies. We formulate a MILP solver as the environment and the HEM as the agent. Moreover, we train HEM via a hierarchical policy gradient algorithm.

based on the information of $(M_{\text{LP}}, \mathcal{C}, \mathbf{x}_{\text{LP}}^*)$. That is, we actually represent a state $s$ by *a sequence of thirteen-dimensional feature vectors*. We present details of the designed features in Appendix F.1. **(2) The action space $\mathcal{A}$.** To take into account the ratio and order of selected cuts, we define the action space by *all the ordered subsets* of the candidate cuts $\mathcal{C}$. It can be challenging to explore the action space efficiently, as the cardinality of the action space can be extremely large due to its combinatorial structure. **(3) The reward function $r$.** To evaluate the impact of the added cuts on solving MILPs, we design the reward function by (i) measures collected at the end of solving LP relaxations such as the dual bound improvement, (ii) or end-of-run statistics, such as the solving time and the primal-dual gap integral. For the first, the reward $r(s, a)$ can be defined as the negative dual bound improvement at each step. For the second, the reward $r(s, a)$ can be defined as zero except for the last step $(s_T, a_T)$ in a trajectory, i.e., $r(s_T, a_T)$ is defined by the negative solving time or the negative primal-dual gap integral. **(4) The transition function $f$.** The transition function maps the current state $s$ and the action $a$ to the next state $s'$, where $s'$ represents the next LP relaxation generated by adding the selected cuts at the current LP relaxation. **(5) The terminal state.** There is no standard and unified criterion to determine when to terminate the cut separation procedure (Paulus et al., 2022). Suppose we set the cut separation rounds as $T$, then the solver environment terminates the cut separation after $T$ rounds. Under the multiple rounds setting (i.e., $T > 1$), we formulate the cut selection as a Markov decision process. Under the one round setting (i.e., $T = 1$), the formulation can be simplified as a contextual bandit.

**Hierarchical Sequence Model**

**Motivation.** Let $\pi$ denote the cut selection policy $\pi : \mathcal{S} \to \mathcal{P}(\mathcal{A})$, where $\mathcal{P}(\mathcal{A})$ denotes the probability distribution over the action space, and $\pi(\cdot|s)$ denotes the probability distribution over the action space given the state $s$. We emphasize that learning such policies can tackle **(P1)-(P3)** in cut selection simultaneously. However, directly learning such policies is challenging for the following reasons. First, it is challenging to explore the action space efficiently, as the cardinality of the action space can be extremely large due to its combinatorial structure. Second, the length and max length of actions (i.e., ordered subsets) are variable across different MILPs. However, traditional RL usually deals with problems whose actions have a fixed length. Instead of directly learning the aforementioned policy, many existing learning-based methods (Tang et al., 2020; Huang et al., 2022; Paulus et al., 2022) learn a scoring function that outputs a score given a cut, and select a fixed ratio/number of cuts with high scores. However, they suffer from two limitations as mentioned in Section 1.

**Policy network architecture.** To tackle the aforementioned problems, we propose a novel hierarchical sequence model (HEM) to learn cut selection policies. To promote efficient exploration, HEM leverages the hierarchical structure of the cut selection task to decompose the policy into two sub-policies, i.e., a higher-level policy $\pi^h$ and a lower-level policy $\pi^l$. The policy network architecture of HEM is also illustrated in Figure 2. **First**, the higher-level policy learns the number of cuts that should be selected by predicting a proper ratio. Suppose the length of the state is $N$ and the predicted ratio is $k$, then the predicted number of cuts that should be selected is $\lfloor N * k \rfloor$, where $\lfloor \cdot \rfloor$ denotes the floor function. We define the higher-level policy by $\pi^h : \mathcal{S} \to \mathcal{P}([0,1])$, where $\pi^h(\cdot|s)$ denotes the probability distribution over $[0,1]$ given the state $s$. **Second**, the lower-level policy learns to select an ordered subset with the size determined by the higher-level policy. We define the lower-level policy by $\pi^l : \mathcal{S} \times [0,1] \to \mathcal{P}(\mathcal{A})$, where $\pi^l(\cdot|s,k)$ denotes the probability distribution over the action space given the state $s$ and the ratio $k$. Specifically, we formulate the lower-level policy as a sequence model, which can capture the interaction among cuts. **Finally**, we derive the cut selection policy via the law of total probability, i.e., $\pi(a_k|s) = \mathbb{E}_{k \sim \pi^h(\cdot|s)}[\pi^l(a_k|s,k)]$, where $k$ denotes the given ratio and $a_k$ denotes the action. The policy is computed by an expectation, as $a_k$ cannot determine the ratio $k$. For example, suppose that $N = 100$ and the length of $a_k$ is 10, then the ratio $k$ can be any number in the interval $[0.1, 0.11)$. Actually, we sample an action from the policy $\pi$ by first sampling a ratio $k$ from $\pi^h$ and then sampling an action from $\pi^l$ given the ratio.

For the higher-level policy, we first model the higher-level policy as a tanh-Gaussian, i.e., a Gaussian distribution with an invertible squashing function ($\tanh$), which is commonly used in deep reinforcement learning (Schulman et al., 2017; Haarnoja et al., 2018). The mean and variance of the Gaussian are given by neural networks. The support of the tanh-Gaussian is $[-1, 1]$, but a ratio of selected cuts should belong to $[0, 1]$. Thus, we further perform a linear transformation on the tanh-Gaussian. Specifically, we define the parameterized higher-level policy by $\pi^h_{\theta_1}(\cdot|s) = 0.5 * \tanh(K) + 0.5$, where $K \sim \mathcal{N}(\mu_{\theta_1}(s), \sigma_{\theta_1}(s))$. Since the sequence lengths of states are variable across different instances (MILPs), we use a long-short term memory (LSTM) (Hochreiter & Schmidhuber, 1997) network to embed the sequence of candidate cuts. We then use a multi-layer perceptron (MLP) (Goodfellow et al., 2016) to predict the mean and variance from the last hidden state of the LSTM.

For the lower-level policy, we formulate it as a sequence model. That is, its input is a sequence of candidate cuts, and its output is the probability distribution over ordered subsets of candidate cuts with the size determined by the higher-level policy. Specifically, given a state action pair $(s, k, a_k)$, the sequence model computes the conditional probability $\pi^l_{\theta_2}(a_k|s,k)$ using a parametric model to estimate the terms of the probability chain rule, i.e., $\pi^l_{\theta_2}(a_k|s,k) = \prod_{i=1}^{m} p_{\theta_2}(a_k^i|a_k^1, \ldots, a_k^{i-1}, s, k)$. Here $s = \{s^1, \ldots, s^N\}$ is the input sequence, $m = \lfloor N * k \rfloor$ is the length of the output sequence, and $a_k = \{a_k^1, \ldots, a_k^m\}$ is a sequence of $m$ indices, each corresponding a position in the input sequence $s$. Such policy can be parametrized by the vanilla sequence model commonly used in machine translation (Sutskever et al., 2014; Vaswani et al., 2017). However, the vanilla sequence model can only be applied to learning on a single instance, as the number of candidate cuts varies on different instances. To generalize across different instances, we use a pointer network (Vinyals et al., 2015; Bello* et al., 2017)—which uses attention as a pointer to select a member of the input sequence as the output at each decoder step—to parametrize $\pi^l_{\theta_2}$ (see Appendix F.4.1 for details). To the best of our knowledge, we are the first to formulate the cut selection task as a sequence to sequence learning problem and apply the pointer network to cut selection. This leads to two major advantages: (1) capturing the underlying order information, (2) and the interaction among cuts. This is also illustrated through an example in Appendix E.

**Training: hierarchical policy gradient**

For the cut selection task, we aim to find $\theta$ that maximizes the expected reward over all trajectories

$$J(\theta) = \mathbb{E}_{s \sim \mu, a_k \sim \pi_\theta(\cdot|s)}[r(s, a_k)], \tag{3}$$

where $\theta = [\theta_1, \theta_2]$ with $[\theta_1, \theta_2]$ denoting the concatenation of the two vectors, $\pi_\theta(a_k|s) = \mathbb{E}_{k \sim \pi^h_{\theta_1}(\cdot|s)}[\pi^l_{\theta_2}(a_k|s, k)]$, and $\mu$ denotes the initial state distribution.

To train the policy with a hierarchical structure, we derive a hierarchical policy gradient following the well-known policy gradient theorem (Sutton et al., 1999a; Sutton & Barto, 2018).

**Proposition 1.** *Given the cut selection policy* $\pi_\theta(a_k|s) = \mathbb{E}_{k \sim \pi^h_{\theta_1}(\cdot|s)}[\pi^l_{\theta_2}(a_k|s, k)]$ *and the training objective (3), the hierarchical policy gradient takes the form of*

$$\nabla_{\theta_1} J([\theta_1, \theta_2]) = \mathbb{E}_{s \sim \mu, k \sim \pi^h_{\theta_1}(\cdot|s)}[\nabla_{\theta_1} \log(\pi^h_{\theta_1}(k|s))\mathbb{E}_{a_k \sim \pi^l_{\theta_2}(\cdot|s,k)}[r(s, a_k)]],$$

$$\nabla_{\theta_2} J([\theta_1, \theta_2]) = \mathbb{E}_{s \sim \mu, k \sim \pi^h_{\theta_1}(\cdot|s), a_k \sim \pi^l_{\theta_2}(\cdot|s,k)}[\nabla_{\theta_2} \log \pi^l_{\theta_2}(a_k|s, k)r(s, a_k)].$$

We provide detailed proof in Appendix A. We use the derived hierarchical policy gradient to update the parameters of the higher-level and lower-level policies. We implement the training algorithm in a parallel manner that is closely related to the asynchronous advantage actor-critic (A3C) (Mnih et al., 2016). Due to limited space, we summarize the procedure of the training algorithm in Appendix F.3.6. Moreover, we discuss some more advantages of HEM (see Appendix F.4.3 for details). (1) HEM leverages the hierarchical structure of the cut selection task, which is important for efficient exploration in complex decision-making tasks (Sutton et al., 1999b). (2) We train HEM via gradient-based algorithms, which is sample efficient (Sutton & Barto, 2018).

## 5 Experiments

Our experiments have five main parts: **Experiment 1.** Evaluate our approach on three classical MILP problems and six challenging MILP problem benchmarks from diverse application areas. **Experiment 2.** Perform carefully designed ablation studies to provide further insight into HEM. **Experiment 3.** Test whether HEM can generalize to instances significantly larger than those seen during training. **Experiment 4.** Visualize the cuts selected by our method compared to the baselines. **Experiment 5.** Deploy our approach to real-world production planning problems.

**Benchmarks.** We evaluate our approach on nine $\mathcal{NP}$-hard MILP problem benchmarks, which consist of three classical synthetic MILP problems and six challenging MILP problems from diverse application areas. We divide the nine problem benchmarks into three categories according to the difficulty of solving them using the SCIP 8.0.0 solver (Bestuzheva et al., 2021). We call the three categories easy, medium, and hard datasets, respectively. (1) *Easy datasets* comprise three widely used synthetic MILP problem benchmarks: Set Covering (Balas & Ho, 1980), Maximum Independent Set (Bergman et al., 2016), and Multiple Knapsack (Scavuzzo et al., 2022). We artificially generate instances following Gasse et al. (2019); Sun et al. (2020). (2) *Medium datasets* comprise MIK (Atamtürk, 2003) and CORLAT (Gomes et al., 2008), which are widely used benchmarks for evaluating MILP solvers (He et al., 2014; Nair et al., 2020). (3) *Hard datasets* include the Load Balancing problem, inspired by real-life applications of large-scale systems, and the Anonymous problem, inspired by a large-scale industrial application (Bowly et al., 2021). Moreover, hard datasets contain benchmarks from MIPLIB 2017 (MIPLIB) (Gleixner et al., 2021). Although Turner et al. (2022) has shown that directly learning over the full MIPLIB can be extremely challenging, we propose to learn over subsets of MIPLIB. We construct two subsets, called MIPLIB mixed neos and MIPLIB mixed supportcase. Due to limited space, please see Appendix D.1 for details of these datasets.

**Experimental setup.** Throughout all experiments, we use SCIP 8.0.0 (Bestuzheva et al., 2021) as the backend solver, which is the state-of-the-art open source solver, and is widely used in research of machine learning for combinatorial optimization (Gasse et al., 2019; Huang et al., 2022; Turner et al., 2022; Nair et al., 2020). Following Gasse et al. (2019); Huang et al. (2022); Paulus et al. (2022), we only allow cutting plane generation and selection at the root node, and set the cut separation rounds as one. We keep all the other SCIP parameters to default so as to make comparisons as fair and reproducible as possible. We emphasize that all of the SCIP solver's advanced features, such as presolve and heuristics, are open, which ensures that our setup is consistent with the practice setting. Throughout all experiments, we set the solving time limit as 300 seconds. For completeness,

Table 1: Policy evaluation on the easy, medium, and hard datasets. The best performance is marked in bold. Let $m$ denote the average number of constraints and $n$ denote the average number of variables. We report the arithmetic mean (standard deviation) of the Time and PD integral.

| | Easy: Set Covering ($n=1000$, $m=500$) | | | Easy: Maximum Independent Set ($n=500$, $m=1953$) | | | Easy: Multiple Knapsack ($n=720$, $m=72$) | | |
|---|---|---|---|---|---|---|---|---|---|
| Method | Time(s)↓ | Improvement (Time, %)↑ | PD integral↓ | Time(s)↓ | Improvement (Time, %)↑ | PD integral↓ | Time(s)↓ | Improvement (Time, %)↑ | PD integral↓ |
| NoCuts | 6.31 (4.61) | NA | 56.99 (38.89) | 8.78 (6.66) | NA | 71.31 (51.74) | 9.88 (22.24) | NA | 16.41 (14.16) |
| Default | 4.41 (5.12) | 29.90 | 55.63 (42.21) | 3.88 (5.04) | 55.80 | 29.44 (35.27) | 9.90 (22.24) | -0.20 | 16.46 (14.25) |
| Random | 5.74 (5.19) | 8.90 | 67.08 (46.58) | 6.50 (7.09) | 26.00 | 52.46 (53.10) | 13.10 (35.51) | -32.60 | 20.00 (25.14) |
| NV | 9.86 (5.43) | -56.50 | 99.77 (53.12) | 7.84 (5.54) | 10.70 | 61.60 (43.95) | 13.04 (36.91) | -32.00 | 21.75 (24.71) |
| Eff | 9.65 (5.45) | -53.20 | 95.66 (51.71) | 7.80 (5.11) | 11.10 | 61.04 (41.88) | 9.99 (19.02) | -1.10 | 20.49 (22.11) |
| SBP | 1.91 (0.36) | 69.60 | 38.96 (8.66) | 2.43 (5.55) | 72.30 | 21.99 (40.86) | 7.74 (12.36) | 21.60 | 16.45 (16.62) |
| HEM (Ours) | **1.85 (0.31)** | **70.60** | **37.92 (8.46)** | **1.76 (3.69)** | **80.00** | **16.01 (26.21)** | **6.13 (9.61)** | **38.00** | **13.63 (9.63)** |

| | Medium: MIK ($n=413$, $m=346$) | | | Medium: Corlat ($n=466$, $m=486$) | | | Hard: Load Balancing ($n=61000$, $m=64304$) | | |
|---|---|---|---|---|---|---|---|---|---|
| Method | Time(s)↓ | PD integral↓ | Improvement↑ (PD integral, %) | Time(s)↓ | PD integral↓ | Improvement↑ (PD integral, %) | Time(s)↓ | PD integral↓ | Improvement↑ (PD integral, %) |
| NoCuts | 300.01 (0.009) | 2355.87 (996.08) | NA | 103.30 (128.14) | 2818.40 (5908.31) | NA | 300.00 (0.12) | 14853.77 (951.42) | NA |
| Default | 179.62 (122.36) | 844.40 (924.30) | 64.10 | 75.20 (120.30) | 2412.09 (5892.88) | 14.40 | 300.00 (0.06) | 9589.19 (1012.95) | 35.40 |
| Random | 289.86 (28.90) | 2036.80 (933.17) | 13.50 | 84.18 (124.34) | 2501.98 (6031.43) | 11.20 | 300.00 (0.09) | 13621.20 (1162.02) | 8.30 |
| NV | 299.76 (1.32) | 2542.67 (529.49) | -7.90 | 90.26 (128.33) | 3075.70 (7029.55) | -9.10 | 300.00 (0.05) | 13933.88 (971.10) | 6.20 |
| Eff | 298.48 (5.84) | 2416.57 (642.41) | -2.60 | 104.38 (131.61) | 3155.03 (7039.99) | -11.90 | 300.00 (0.07) | 13913.07 (969.95) | 6.30 |
| SBP | 286.07 (41.81) | 2053.30 (740.11) | 12.80 | 70.41 (122.17) | 2023.87 (5085.96) | 28.20 | 300.00 (0.10) | 12535.30 (741.43) | 15.60 |
| HEM (Ours) | **176.12 (125.18)** | **785.04 (790.38)** | **66.70** | **58.31 (110.51)** | **1079.99 (2653.14)** | **61.68** | **300.00 (0.04)** | **9496.42 (1018.35)** | **36.10** |

| | Hard: Anonymous ($n=37881$, $m=49603$) | | | Hard: MIPLIB mixed neos ($n=6958$, $m=5660$) | | | Hard: MIPLIB mixed supportcase ($n=19766$, $m=19910$) | | |
|---|---|---|---|---|---|---|---|---|---|
| Method | Time(s)↓ | PD integral↓ | Improvement↑ (PD integral, %) | Time(s)↓ | PD integral↓ | Improvement↑ (PD integral, %) | Time(s)↓ | PD integral↓ | Improvement↑ (PD integral, %) |
| NoCuts | 246.22 (94.90) | 18297.30 (9769.42) | NA | 253.65 (80.29) | 14652.29 (12523.37) | NA | 170.00 (131.60) | 9927.96 (11334.07) | NA |
| Default | 244.02 (97.72) | 17407.01 (9736.19) | 4.90 | 256.58 (76.05) | 14444.05 (12347.09) | 1.42 | 164.61 (135.82) | 9672.34 (10668.24) | 2.57 |
| Random | 243.49 (98.21) | 16850.89 (10227.87) | 7.80 | 255.88 (76.65) | 14006.48 (12698.76) | 4.41 | 165.88 (134.40) | 10034.70 (11052.73) | -1.07 |
| NV | 242.01 (98.68) | 16873.66 (9711.16) | 7.80 | 263.81 (64.10) | 14379.05 (12306.35) | 1.86 | 161.67 (131.43) | 8967.00 (9690.30) | 9.68 |
| Eff | 244.94 (93.47) | 17137.87 (9456.34) | 6.30 | 260.53 (68.54) | 14021.74 (12859.41) | 4.30 | 167.35 (134.99) | 9941.55 (10943.48) | -0.14 |
| SBP | 245.71 (92.46) | 18188.63 (9651.85) | 0.59 | 256.48 (78.59) | 13531.00 (12898.22) | 7.65 | 165.61 (135.25) | 7408.65 (7903.47) | 25.37 |
| HEM (Ours) | **241.68 (97.23)** | **16077.15 (9108.21)** | **12.10** | **248.66 (89.46)** | **8678.76 (12337.00)** | **40.77** | **162.96 (138.21)** | **6874.80 (6729.97)** | **30.75** |

we also evaluate HEM with a much longer time limit of three hours. The results are given in Appendix G.6. We train HEM with ADAM (Kingma & Ba, 2014) using the PyTorch (Paszke et al., 2019). Additionally, we also provide another implementation using the MindSpore (Chen, 2021). For simplicity, we split each dataset into the train and test sets with $80\%$ and $20\%$ instances. To further improve HEM, one can construct a valid set for hyperparameters tuning. We train our model on the train set, and select the best model on the train set to evaluate on the test set. Please refer to Appendix F.3 for implementation details, hyperparameters, and hardware specification.

**Baselines.** Our baselines include five widely used human-designed cut selection rules and a state-of-the-art (SOTA) learning-based method. Cut selection rules include NoCuts, Random, Normalized Violation (NV), Efficacy (Eff), and Default. NoCuts does not add any cuts. Default denotes the default cut selection rule used in SCIP 8.0.0. For learning-based methods, we implement a *slight* variant of the SOTA learning-based methods (Tang et al., 2020; Huang et al., 2022), namely score-based policy (SBP). Please see Appendix F.2 for implementation details of these baselines.

**Evaluation metrics.** We use two widely used evaluation metrics, i.e., the average solving time (Time, lower is better), and the average primal-dual gap integral (PD integral, lower is better). Additionally, we provide more results in terms of another two metrics, i.e., the average number of nodes and the average primal-dual gap, in Appendix G.2. Furthermore, to evaluate different cut selection methods compared to pure branch-and-bound without cutting plane separation, we propose an *Improvement* metric. Specifically, we define the metric by $\text{Im}_M(\cdot) = \frac{M(\text{NoCuts}) - M(\cdot)}{M(\text{NoCuts})}$, where $M(\text{NoCuts})$ represents the performance of NoCuts, and $M(\cdot)$ represents a mapping from a method to its performance. The improvement metric represents the improvement of a given method compared to NoCuts. *We mainly focus on the Time metric on the easy datasets*, as the solver can solve all instances to optimality within the given time limit. However, HEM and the baselines cannot solve all instances to optimality within the time limit on the medium and hard datasets. As a result, the average solving time of those unsolved instances is the same, which makes it difficult to distinguish the performance of different cut selection methods using the Time metric. Therefore, *we mainly focus on the PD integral metric on the medium and hard datasets*. The PD integral is also a well-recognized metric for evaluating the solver performance (Bowly et al., 2021; Cao et al., 2022).

**Experiment 1. Comparative evaluation** The results in Table 1 suggest the following. (1) **Easy datasets.** HEM significantly outperforms all the baselines on the easy datasets, especially on Maximum Independent Set and Multiple Knapsack. SBP achieves much better performance than all the rule-based baselines, demonstrating that our implemented SBP is a strong baseline. Compared to SBP, HEM improves the Time by up to $16.4\%$ on the three datasets, demonstrating the superiority of our method over the SOTA learning-based method. (2) **Medium datasets.** On MIK and CORLAT,

Table 2: Comparison between HEM and HEM without the higher-level model.

| | Easy: Maximum Independent Set ($n = 500$, $m = 1953$) | | | Medium: Corlat ($n = 466$, $m = 486$) | | | Hard: MIPLIB mixed neos ($n = 6958$, $m = 5660$) | | |
|---|---|---|---|---|---|---|---|---|---|
| Method | Time(s) ↓ | Improvement ↑ (Time, %) | PD integral ↓ | Time(s) ↓ | PD integral ↓ | Improvement ↑ (PD integral, %) | Time(s) ↓ | PD integral ↓ | Improvement ↑ (PD integral, %) |
| NoCuts | 8.78 (6.66) | NA | 71.31 (51.74) | 103.30 (128.14) | 2818.40 (5908.31) | NA | 253.65 (80.29) | 14652.29 (12523.37) | NA |
| Default | 3.88 (5.04) | 55.81 | 29.44 (35.27) | 75.20 (120.30) | 2412.09 (5892.88) | 14.42 | 256.58 (76.05) | 14444.05 (12347.09) | 1.42 |
| SBP | 2.43 (5.55) | 72.32 | 21.99 (40.86) | 70.41 (122.17) | 2023.87 (5085.96) | 28.19 | 256.48 (78.59) | 13531.00 (12898.22) | 7.65 |
| HEM w/o H | 1.88 (4.20) | 78.59 | 16.70 (28.15) | 63.14 (115.36) | 1939.08 (5484.83) | 31.20 | 249.21 (88.09) | 13614.29 (12914.76) | 7.08 |
| HEM (Ours) | **1.76 (3.69)** | **79.95** | **16.01 (26.21)** | **58.31 (110.51)** | **1079.99 (2653.14)** | **61.68** | 248.66 (89.46) | **8678.76 (12337.00)** | **40.77** |

Table 3: Comparison between HEM, HEM-ratio, and HEM-ratio-order.

| | Easy: Maximum Independent Set ($n = 500$, $m = 1953$) | | | Medium: Corlat ($n = 466$, $m = 486$) | | | Hard: MIPLIB mixed neos ($n = 6958$, $m = 5660$) | | |
|---|---|---|---|---|---|---|---|---|---|
| Method | Time(s) ↓ | Improvement ↑ (Time, %) | PD integral ↓ | Time(s) ↓ | PD integral ↓ | Improvement ↑ (PD integral, %) | Time(s) ↓ | PD integral ↓ | Improvement ↑ (PD integral, %) |
| NoCuts | 8.78 (6.66) | NA | 71.31 (51.74) | 103.30 (128.14) | 2818.40 (5908.31) | NA | 253.65 (80.29) | 14652.29 (12523.37) | NA |
| Default | 3.88 (5.04) | 55.81 | 29.44 (35.27) | 75.20 (120.30) | 2412.09 (5892.88) | 14.42 | 256.58 (76.05) | 14444.05 (12347.09) | 1.42 |
| SBP | 2.43 (5.55) | 72.32 | 21.99 (40.86) | 70.41 (122.17) | 2023.87 (5085.96) | 28.19 | 256.48 (78.59) | 13531.00 (12898.22) | 7.65 |
| HEM-ratio-order | 2.30 (5.18) | 73.80 | 21.19 (38.52) | 70.94 (122.93) | 1416.66 (3380.10) | 49.74 | 245.99 (93.67) | 14026.75 (12683.60) | 4.27 |
| HEM-ratio | 2.26 (5.06) | 74.26 | 20.82 (37.80) | 67.27 (117.01) | 1251.60 (2869.87) | 55.59 | **244.87 (95.56)** | 13659.93 (12900.59) | 6.77 |
| HEM (Ours) | **1.76 (3.69)** | **79.95** | **16.01 (26.21)** | **58.31 (110.51)** | **1079.99 (2653.14)** | **61.68** | 248.66 (89.46) | **8678.76 (12337.00)** | **40.77** |

HEM still outperforms all the baselines. Especially on CORLAT, HEM achieves at least $33.48\%$ improvement in terms of the PD integral compared to the baselines. (3) **Hard datasets.** HEM significantly outperforms the baselines in terms of the PD integral on several problems in the hard datasets. HEM achieves outstanding performance on two challenging datasets from MIPLIB 2017 and real-world problems (Load Balancing and Anonymous), demonstrating the powerful ability to enhance MILP solvers with HEM in large-scale real-world applications. Moreover, SBP performs extremely poorly on several medium and hard datasets, which implies that it can be difficult to learn good cut selection policies on challenging MILP problems.

**Experiment 2. Ablation study** We present ablation studies on Maximum Independent Set (MIS), CORLAT, and MIPLIB mixed neos, which are representative datasets from the easy, medium, and hard datasets. We provide more results on the other datasets in Appendix G.3.

**Contribution of each component.** We perform ablation studies to understand the contribution of each component in HEM. We report the performance of HEM and HEM without the higher-level model (HEM w/o H) in Table 2. HEM w/o H is essentially a pointer network. Note that it can still implicitly predicts the number of cuts that should be selected by predicting an end token as used in language tasks (Sutskever et al., 2014). Please see Appendix F.4.2 for details. **First**, the results in Table 2 show that HEM w/o H outperforms all the baselines on MIS and CORLAT, demonstrating the advantages of the lower-level model. Although HEM w/o H outperforms Default on MIPLIB mixed neos, HEM w/o H performs on par with SBP. A possible reason is that it is difficult for HEM w/o H to explore the action space efficiently, and thus HEM w/o H tends to be trapped to the local optimum. **Second**, the results in Table 2 show that HEM significantly outperforms HEM w/o H and the baselines on the three datasets. The results demonstrate that the higher-level model is important for efficient exploration in complex tasks, thus significantly improving the solving efficiency.

**The importance of tackling (P1)-(P3).** We perform ablation studies to understand the importance of tackling **(P1)-(P3)** in cut selection. (1) **HEM.** HEM tackles **(P1)-(P3)** in cut selection simultaneously. (2) **HEM-ratio.** In order not to learn how many cuts should be selected, we remove the higher-level model of HEM and *force the lower-level model to select a fixed ratio of cuts*. We denote it by HEM-ratio. Note that HEM-ratio is different from HEM w/o H (see Appendix F.4.2). HEM-ratio tackles **(P1)** and **(P3)** in cut selection. (3) **HEM-ratio-order.** To further mute the effect of the order of selected cuts, we reorder the selected cuts given by HEM-ratio with the original index of the generated cuts, which we denote by HEM-ratio-order. HEM-ratio-order mainly tackles **(P1)** in cut selection. The results in Table 3 suggest the following. HEM-ratio-order significantly outperforms Default and NoCuts, demonstrating that tackling **(P1)** by data-driven methods is crucial. HEM significantly outperforms HEM-ratio in terms of the PD integral, demonstrating the significance of tackling **(P2)**. HEM-ratio outperforms HEM-ratio-order in terms of the Time and the PD integral, which demonstrates the importance of tackling **(P3)**. Moreover, HEM-ratio and HEM-ratio-order perform better than SBP on MIS and CORLAT, demonstrating the advantages of using the sequence model to learn cut selection over SBP. HEM-ratio and HEM-ratio-order perform on par with SBP on MIPLIB mixed neos. We provide possible reasons in Appendix G.3.1.

**Experiment 3. Generalization** We evaluate the ability of HEM to generalize across different sizes of MILPs. Let $n \times m$ denote the size of MILP instances. Following Gasse et al. (2019); Sun et al. (2020), we test the generalization ability of HEM on Set Covering and Maximum Independent Set (MIS), as we can artificially generate instances with arbitrary sizes for synthetic MILP problems.

Table 4: **Left**: The generalization ability of HEM. **Right**: Test on Production Planning problems.

| | Maximum Independent Set ($n = 1000$, $m = 3946$, 4×) | | | Maximum Independent Set ($n = 5940$, $m = 1500$, 9×) | | | | Production Planning ($n = 3582.25$, $m = 5040.42$) | | | |
|---|---|---|---|---|---|---|---|---|---|---|---|
| Method | Time(s)↓ | Improvement↑ (Time, %) | PD integral↓ | Time(s)↓ | Improvement↑ (Time, %) | PD integral↓ | Method | Time (s)↓ | Improvement↑ (Time, %) | PD integral↓ | Improvement↑ (PD integral, %) |
| NoCuts | 170.06 (100.61) | NA | 874.45 (522.29) | 300.00 (0) | NA | 2019.93 (353.27) | NoCuts | 278.79 (231.02) | NA | 17866.01 (21309.85) | NA |
| Default | 42.40 (76.00) | 48.72 | 198.61 (331.20) | 111.18 (144.13) | 60.91 | 616.46 (798.94) | Default | 296.12 (246.25) | -6.22 | 17703.39 (21330.40) | 0.91 |
| Random | 118.25 (109.05) | -43.00 | 574.33 (516.11) | 245.13 (115.80) | 13.82 | 1562.20 (793.09) | Random | 280.18 (237.09) | -0.50 | 18120.21 (21660.01) | -1.42 |
| NV | 160.30 (101.41) | -93.86 | 784.98 (493.24) | 299.97 (0.49) | -5.46 | 1922.52 (349.67) | NV | 259.48 (227.81) | 6.93 | 17295.18 (21860.07) | 3.20 |
| Eff | 158.75 (100.40) | -91.98 | 779.63 (493.05) | 299.45 (3.77) | -5.28 | 1921.61 (361.26) | Eff | 263.60 (229.24) | 5.45 | 16636.52 (21322.89) | 6.88 |
| SBP | 50.55 (89.14) | 38.87 | 253.81 (426.94) | 108.42 (143.68) | 61.88 | 680.41 (903.88) | SBP | 276.61 (235.84) | 0.78 | 16952.85 (21386.07) | 5.11 |
| HEM (Ours) | 35.34 (67.91) | 57.26 | 160.56 (282.03) | 108.02 (143.02) | 62.02 | 570.48 (760.65) | HEM (Ours) | 241.77 (229.97) | 13.28 | 15751.08 (20683.53) | 11.84 |

On MIS, we test HEM on four times and nine times larger instances than those seen during training. The results in Table 4 (**Left**) show that HEM significantly outperforms the baselines in terms of the Time and the PD integral on 4× and 9× MIS, demonstrating the superiority of HEM in terms of the generalization ability. Interestingly, SBP also generalizes well to large instances, demonstrating that SBP is a strong baseline. We provide more results on Set Covering in Appendix G.4.

**Experiment 4. Visualization of selected cuts** We visualize the diversity of selected cuts, an important metric for evaluating whether the selected cuts complement each other nicely (Dey & Molinaro, 2018b). We visualize the cuts selected by HEM-ratio and SBP on a randomly sampled instance from Maximum Independent Set and CORLAT, respectively. We evaluate HEM-ratio rather than HEM, as HEM-ratio selects the

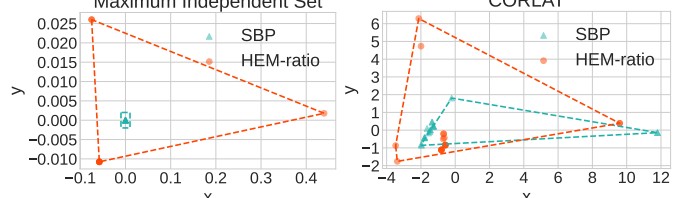

Figure 3: We perform principal component analysis on the cuts selected by HEM-ratio and SBP. Colored points illustrate the reduced cut features. The area covered by the dashed lines represents the diversity of selected cuts. The results show that HEM-ratio selects much more diverse cuts than SBP.

same number of cuts as SBP. Furthermore, we perform principal component analysis on the selected cuts to reduce the cut features to two-dimensional space. Colored points illustrate the reduced cut features. To visualize the diversity of selected cuts, we use dashed lines to connect the points with the smallest and largest x,y coordinates. That is, *the area covered by the dashed lines represents the diversity*. Figure 3 shows that SBP tends to select many similar cuts that are possibly redundant, especially on Maximum Independent Set. In contrast, HEM-ratio selects much more diverse cuts that can well complement each other. Please refer to Appendix G.5 for results on more datasets.

**Experiment 5. Deployment in real-world production planning problems** To further evaluate the effectiveness of our proposed HEM, we deploy HEM to large-scale real-world production planning problems at an anonymous enterprise, which is one of the largest global commercial technology enterprises. Please refer to Appendix D.3 for more details of the problems. The results in Table 4 (**Right**) show that HEM significantly outperforms all the baselines in terms of the Time and PD integral. The results demonstrate the strong ability to enhance modern MILP solvers with our proposed HEM in real-world applications. Interestingly, Default performs poorer than NoCuts, which implies that an improper cut selection policy could significantly degrade the performance of MILP solvers. In addition, we will integrate our proposed HEM into OptVerse[1], i.e., the commercial solver developed by Huawei.

## 6 CONCLUSION

In this paper, we observe from extensive empirical results that the order of selected cuts has a significant impact on the efficiency of solving MILPs. We propose a novel **h**ierarchical s**e**quence **m**odel (HEM) to learn cut selection policies via reinforcement learning. Specifically, HEM consists of a two-level model: (1) a higher-level model to learn the number of cuts that should be selected, (2) and a lower-level model—that formulates the cut selection task as a sequence to sequence learning problem—to learn policies selecting an ordered subset with the size determined by the higher-level model. Experiments show that HEM significantly improves the efficiency of solving MILPs compared to human-designed and learning-based baselines on both synthetic and large-scale real-world MILPs. We believe that our proposed approach brings new insights into learning cut selection.

---

[1]Please refer to `https://www.huaweicloud.com/product/modelarts/optverse.html` for details of OptVerse.

## ACKNOWLEDGEMENTS

The authors would like to thank all the anonymous reviewers for their insightful comments. This work was supported in part by National Nature Science Foundations of China grants U19B2026, U19B2044, 61836011, 62021001, and 61836006, and the Fundamental Research Funds for the Central Universities grant WK3490000004. We gratefully acknowledge the support of MindSpore used for this research. In addition, we would like to gratefully thank all the developers of OptVerse, and Huawei Cloud Solver Lab for their support of this research.

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

## A PROOF

### A.1 PROOF OF PROPOSITION 1

*Proof.* The optimization objective takes the form of

$$J(\theta) = \mathbb{E}_{s\sim\mu, a_k\sim\pi_\theta(\cdot|s)}[r(s, a_k)]$$

$$= \mathbb{E}_{s\sim\mu}[\sum_{a_k} \int_{k=0}^1 \pi_{\theta_1}^h(k|s)\pi_{\theta_2}^l(a_k|s,k)r(s,a_k)dk]$$

$$= \mathbb{E}_{s\sim\mu}[\int_{k=0}^1 \sum_{a_k} \pi_{\theta_1}^h(k|s)\pi_{\theta_2}^l(a_k|s,k)r(s,a_k)dk],$$

where $\theta = [\theta_1, \theta_2]$ with $[\theta_1, \theta_2]$ denoting the concatenation of the two vectors, $\pi_\theta(a_k|s) = \mathbb{E}_{k\sim\pi_{\theta_1}^h(\cdot|s)}[\pi_{\theta_2}^h(a_k|s,k)]$, and $\mu$ denotes the initial state distribution.

We first compute the policy gradient for $\theta_1$:

$$\nabla_{\theta_1} J([\theta_1, \theta_2])$$

$$= \nabla_{\theta_1} (\mathbb{E}_{s\sim\mu}[\int_{k=0}^1 \sum_{a_k} \pi_{\theta_1}^h(k|s)\pi_{\theta_2}^l(a_k|s,k)r(s,a_k)dk])$$

$$= \mathbb{E}_{s\sim\mu}[\nabla_{\theta_1}[\int_{k=0}^1 \pi_{\theta_1}^h(k|s) \sum_{a_k} \pi_{\theta_2}^l(a_k|s,k)r(s,a_k)dk]].$$

Let

$$r^h(s, k, \theta_2) = \sum_{a_k} \pi_{\theta_2}^l(a_k|s,k)r(s,a_k)$$

$$= \mathbb{E}_{a_k\sim\pi_{\theta_2}^l(\cdot|s,k)}[r(s,a_k)],$$

then we have that

$$\nabla_{\theta_1} J([\theta_1, \theta_2]) = \mathbb{E}_{s\sim\mu}[\nabla_{\theta_1}[\int_{k=0}^1 \pi_{\theta_1}^h(k|s)r(s,k,\theta_2)dk]]$$

$$= \mathbb{E}_{s\sim\mu, k\sim\pi_{\theta_1}^h(\cdot|s)}[\nabla_{\theta_1} \log\pi_{\theta_1}^h(k|s)r(s,k,\theta_2)].$$

Therefore, we have that

$$\nabla_{\theta_1} J([\theta_1, \theta_2])$$

$$= \mathbb{E}_{s\sim\mu, k\sim\pi_{\theta_1}^h(\cdot|s)}[\nabla_{\theta_1} \log(\pi_{\theta_1}^h(k|s))\mathbb{E}_{a_k\sim\pi_{\theta_2}^l(\cdot|s,k)}[r(s,a_k)]].$$

We then compute the policy gradient for $\theta_2$:

$$\nabla_{\theta_2} J([\theta_1, \theta_2])$$

$$= \nabla_{\theta_2} (\mathbb{E}_{s\sim\mu}[\int_{k=0}^1 \sum_{a_k} \pi_{\theta_1}^h(k|s)\pi_{\theta_2}^l(a_k|s,k)r(s,a_k)dk])$$

$$= \mathbb{E}_{s\sim\mu, k\sim\pi_{\theta_1}^h(\cdot|s)}[\nabla_{\theta_2}[\sum_{a_k} \pi_{\theta_2}^l(a_k|s,k)r(s,a_k)]]$$

$$= \mathbb{E}_{s\sim\mu, k\sim\pi_{\theta_1}^h(\cdot|s), a_k\sim\pi_{\theta_2}^l(\cdot|s,k)}[\nabla_{\theta_2} \log\pi_{\theta_2}^l(a_k|s,k)r(s,a_k)],$$

which completes the proof. □

## B RELATED WORK

**Machine learning for MILP.** The use of machine learning methods to help improve the MILP solver performance has been an active topic of significant interest in recent years (Bengio et al.,

2021; Lodi & Zarpellon, 2017; Bowly et al., 2021; Gasse et al., 2019; Qu et al., 2022b; Li et al., 2023). During the solving process of the solvers, many crucial decisions that significantly impact the solver performance are based on heuristics (Achterberg, 2007). Recent methods propose to replace these hand-crafted heuristics with machine learning models (Bengio et al., 2021). This line of research has shown significant improvement on the solver performance, including cut selection (Tang et al., 2020; Paulus et al., 2022; Turner et al., 2022; Baltean-Lugojan et al., 2019), variable selection (Khalil et al., 2016; Gasse et al., 2019; Balcan et al., 2018; Zarpellon et al., 2021; Qu et al., 2022a), node selection (He et al., 2014; Sabharwal et al., 2012), column generation (Morabit et al., 2021), and primal heuristics selection (Khalil et al., 2017; Hendel et al., 2019). In this paper, we focus on cut selection, which plays a significant role in modern MILP solvers (Dey & Molinaro, 2018a; Tang et al., 2020).

For cut selection, many existing learning-based methods (Tang et al., 2020; Paulus et al., 2022; Huang et al., 2022) focus on learning which cuts should be preferred by learning a scoring function to measure cut quality. Specifically, (Tang et al., 2020) proposes a reinforcement learning approach to learn to score Gomory cuts (Gomory, 1960) and select a Gomory cut with the best scores. Furthermore, (Paulus et al., 2022) designs a lookahead selection rule which selects a cut that yields the best dual bound improvement, and proposes to learn the expert rule via imitation learning. Instead of selecting the best cut, (Huang et al., 2022) frames cut selection as multiple instance learning to learn a scoring function and selects a fixed ratio of cuts with high scores. However, they neglect the importance of learning how many cuts should be selected. Moreover, we empirically show that the *order of selected cuts* has a large impact on the efficiency of solving MILPs (see Section 3).

Moreover, (Turner et al., 2022) proposes to learn the weightings of four existing scoring rules designed by experts. For the theoretical analysis, (Balcan et al., 2021) provides some provable guarantees for learning cut selection policies.

**Sequence model.** Sequence model such as long-short term memory and Transformer has achieved outstanding performance in language tasks such as machine translation (Hochreiter & Schmidhuber, 1997; Sutskever et al., 2014; Vaswani et al., 2017). For combinatorial optimization, recent works (Vinyals et al., 2015; Bello* et al., 2017) propose a variant of the traditional sequence model, namely pointer network, which is applied to directly finding solutions for specific combinatorial optimization problems, such as the Travelling Salesman Problem (Lenstra & Shmoys, 2009). Instead of finding solutions directly, we propose to use the pointer network for cut selection in modern MILP solvers. To the best of our knowledge, we are the first to apply the pointer network to cut selection, which not only captures the order of selected cuts, but also can well capture the interaction among cuts to select cuts that complement each other nicely.

**Reinforcement learning.** Reinforcement learning (RL) has achieved great success in decision-making tasks, ranging from playing video games (Mnih et al., 2015; Fan, 2021; Fan & Xiao, 2022) to controlling robots in simulators (Haarnoja et al., 2018; Yang et al., 2022). Roughly speaking, RL approaches fall into two categories: (1) model-based RL methods (Janner et al., 2019; Wang et al., 2022b; Zhou et al., 2020), and (2) model-free methods (Haarnoja et al., 2018; Wang et al., 2022a; Kuang et al., 2022; Fan et al., 2021). In this paper, we propose a novel RL framework for learning cut selection policies.

## C MORE DETAILS OF BACKGROUND

### C.1 MORE DETAILS OF THE PRIMAL-DUAL GAP INTEGRAL

We keep track of two important bounds when running branch-and-cut, including the global primal and dual bound. The global primal bound corresponds to the value of the best feasible solution found so far, which is the best upper bound of the problem in (1). The global dual bound corresponds to the minimum dual bound across all leaves of the search tree, which is the best lower bound of the problem in (1). We define the *primal-dual gap integral* by the area between the curve of the solver's global primal bound and the curve of the solver's global dual bound. With a time limit $T$, we define the primal-dual gap integral by

$$\int_{t=0}^{T} (\mathbf{c}^T \mathbf{x}_t^* - \mathbf{z}_t^*) \mathrm{d}t,$$

---

**Algorithm 1** Pseudo code for constructing MIPLIB datasets

---

1: **Input** the initial instance $I_0$, the set of full MIPLIB $\mathcal{M}$, an empty set $\mathcal{M}'$, an empty queue $Q$.
2: Initialize $Q$ with the instance $I_0$, $I_0 \to Q$
3: **while** Q is not empty **do**
4:     n=Q.size()
5:     **for** $i = 1, \ldots, n$ **do**
6:         Pull an element from $Q$, namely $I'$
7:         Compute the similarity scores between each instance in $\mathcal{M}$ except $I'$ and $I'$
8:         Select five instances with the best similarity scores $\mathcal{M}_i$
9:         **for** $I$ in $\mathcal{M}_i$ **do**
10:             **if** $I$ not in $\mathcal{M}'$ **then**
11:                 Push $I$ to $\mathcal{M}'$; Push $I$ to $Q$
12:             **end if**
13:         **end for**
14:     **end for**
15: **end while**
16: **Return** $\mathcal{M}'$

---

where $\mathbf{c}$ is the objective coefficient vector as in (1), $\mathbf{x}_t^*$ is the best feasible solution found at time $t$, $\mathbf{z}_t^*$ is the best dual bound at time $t$. We define the *primal-dual gap* by the difference between the global primal bound and the global dual bound. In SCIP 8.0.0 (Bestuzheva et al., 2021), the initial value of the primal-dual gap is set to a constant 100. The primal-dual gap integral is a well-recognized metric for evaluating solver performance. For example, the primal-dual gal integral is a primary evaluation metric in the NeurIPS 2021 ML4CO competition (Bowly et al., 2021).

## D    DETAILS OF THE DATASETS USED IN THIS PAPER

### D.1    THE DATASETS USED IN THE MAIN EVALUATION

**Easy datasets.** The SCIP 8.0.0 solver needs one minute to solve the MILP instances in the easy datasets to optimality. Easy datasets are comprised of three synthetic MILP problems: Set Covering (Balas & Ho, 1980), Maximum Independent Set (Bergman et al., 2016), and Multiple Knapsack (Scavuzzo et al., 2022). We choose these three classes of problems for the following reasons. First, they are widely used benchmarks for evaluating MILP solvers (Gasse et al., 2019; Huang et al., 2022; Sun et al., 2020; Gupta et al., 2022). Second, they represent a wide collection of MILP problems encountered in practice. Third, for each class of these problems, the average number of generated cuts is at least twenty, which ensures that proper cut selection strategies are significant for improving the solver performance. Similarly to (Gasse et al., 2019; Scavuzzo et al., 2022; Sun et al., 2020; Gupta et al., 2022), we generate set covering instances with 500 rows and 1000 columns, Maximum Independent Set instances with graphs of 500 nodes and affinity set to 4, multiple knapsack instances with 60 items and 12 knapsacks. For each benchmark, we generate a training set of 10,000 instances, and a test set of 100 instances that are never seen during training. Specifically, readers can refer to `https://github.com/ds4dm/learn2branch` or `https://github.com/lascavana/rl2branch` for code to generate the easy datasets. We will also release our code once the paper is accepted to be published.

**Medium datasets.** The SCIP 8.0.0 solver needs at least five minutes to solve the instances in the medium datasets to optimality. Following He et al. (2014); Hutter et al. (2010); Nair et al. (2020), medium datasets comprise MIK (Atamtürk, 2003), a set of MILP problems with knapsack constraints, and CORLAT (Gomes et al., 2008), a real dataset used for the construction of a wildlife corridor for grizzly bears in the Northern Rockies region. Each problem set is split into training and test sets with 80% and 20% of the instances. Readers can refer to `https://atamturk.ieor.berkeley.edu/data/mixed.integer.knapsack/` for MIK. Readers can refer to `https://bitbucket.org/mlindauer/aclib2/src/master/` for CORLAT.

**Hard datasets.** The SCIP 8.0.0 solver needs at least one hour to solve the instances in the hard datasets to optimality.

Table 5: Criteria for removing instances from MIPLIB 2017.

| Criteria | % of instances removed |
|---|---|
| Tags: *feasibility, numerics, infeasible, no solution* | 4.5%, 17.4%, 2.8%, 0.9% |
| Presolve longer than 300 seconds under default conditions | 4.8% |
| Solved to optimality at root | 9.9% |

Table 6: The statistical description of used datasets. In all datasets, $m$ denotes the average number of constraints and $n$ denotes the average number of variables. Inference Time denotes the inference time of our proposed HEM given the average number of candidate cuts.

| Datasets | Set Covering | Maximum Independent Set | Multiple Knapsack | MIK | CORLAT | Load Balancing | Anonymous | MIPLIB mixed neos | MIPLIB mixed supportcase |
|---|---|---|---|---|---|---|---|---|---|
| $m$ | 500 | 1953 | 72 | 346 | 486 | 64304 | 49603 | 5660 | 19910 |
| $n$ | 1000 | 500 | 720 | 413 | 466 | 61000 | 37881 | 6958 | 19766 |
| Avg. Candidate Cuts | $780.51 \pm 289.92$ | $57.04 \pm 15.53$ | $45.00 \pm 12.71$ | $62.00 \pm 13.1$ | $60.00 \pm 33.29$ | $392.53 \pm 32.92$ | $79.40 \pm 72.64$ | $239.00 \pm 154$ | $173.25 \pm 267.27$ |
| Inference Time (s) | 1.58 | 0.11 | 0.09 | 0.12 | 0.12 | 0.77 | 0.15 | 0.47 | 0.34 |

## (1) Benchmarks from MIPLIB

**2017.** Note that MIPLIB 2017 (MIPLIB) (Gleixner et al., 2021) contains instances of MILPs across many different application areas and has been used as a long-standing standard benchmark for MILP solvers (Nair et al., 2020; Turner et al., 2022; Gleixner et al., 2021). Previous work (Turner et al., 2022) has shown that directly learning over the full MIPLIB can be extremely challenging, as these instances are heterogeneous but machine learning has difficulty in learning from heterogeneous datasets. Despite this challenge, we take the first step towards learning over subsets of MIPLIB. Specifically, we construct two subsets by selecting similar instances from MIPLIB. We measure the similarity between instances by 100 human-designed instance features (Gleixner et al., 2021). Following Turner et al. (2022), we first discard instances from MILLIB that satisfy any of the criteria in Table 5. This ensures that a good cut selection policy can significantly improve the dual bound on the remaining instances. Note that we only use three of seven criteria that are used in (Turner et al., 2022) to preserve as many instances as possible.

To select similar instances from MIPLIB 2017, we first choose a representative instance with knapsack constraints (neos-1456979), and a representative instance with set covering constraints (supportcase40). Then we construct the dataset MIPLIB mixed neos following the procedure in Algorithm 1 with the initial instance neos-1456979. We construct the dataset MIPLIB mixed supportcase following the procedure in Algorithm 1 with the initial instance supportcase40. Note that We measure the similarity between instances by 100 human-designed instance features (Gleixner et al., 2021). Each dataset is split into training and test sets with $80\%$ and $20\%$ of the instances.

Specifically, MIPLIB mixed neos contains 20 instances: neos-1456979, ic97_tension, icir97_tension, l2p12, lectsched-4-obj, lectsched-5-obj, loopha13, neos-686190, neos-2294525-abba, neos-3009394-lami, neos-3046601-motu, neos-3046615-murg, neos-3610173-itata, neos-4338804-snowy, neos-5221106-oparau, neos-5260764-orauea, neos-5261882-treska, neos-5266653-tugela, neos16, and timtab1CUTS.

Moreover, MIPLIB mixed supportcase contains 40 instances: supportcase40, 30_70_45_05_100, 30_70_45_095_100, acc-tight2, acc-tight4, acc-tight5, comp07-2idx, comp08-2idx, comp12-2idx, comp21-2idx, decomp1, decomp2, gus-sch, istanbul-no-cutoff, mkc, mkc1, neos-555343, neos-555424, neos-738098, neos-872648, neos-933562, neos-933638, neos-933966, neos-935234, neos-935769, neos-983171, neos-1330346, neos-1337307, neos-1396125, neos-3209462-rhin, neos-3755335-nizao, neos-3759587-noosa, neos-4300652-rahue, neos18, physiciansched6-1, physiciansched6-2, piperout-d27, qiu, reblock354, and supportcase37.

**(2) Benchmarks used in NeurIPS 2021 ML4CO competition** The Load Balancing and Anonymous problems used in the main text are from the NeurIPS 2021 ML4CO competition (Bowly et al., 2021). Readers can refer to `https://www.ecole.ai/2021/ml4co-competition/` for details of the competition. The competition releases three challenging datasets, but we only use two of the three datasets. The major reason is that the average number of the candidate cuts on the instances from the third dataset (Item Placement) is less than five, which makes cut selection has little impact on the overall solver performance.

### D.1.1 DETAILED DESCRIPTION OF THE AFOREMENTIONED DATASETS

In this part, we provide detailed description of the aforementioned datasets. Note that all datasets we use except MIPLIB 2017 are application-specific, i.e., they contain instances from only a single application. We summarize the statistical description of the used datasets in this paper in Table 6. Let $n, m$ denote the average number of variables and constraints in the MILPs. Let $m \times n$ denote the size of the MILPs. We emphasize that the largest size of our used datasets is up to two orders of magnitude larger than that used in previous learning-based cut selection methods (Tang et al., 2020; Paulus et al., 2022), which demonstrates the superiority of our proposed HEM. Moreover, we test

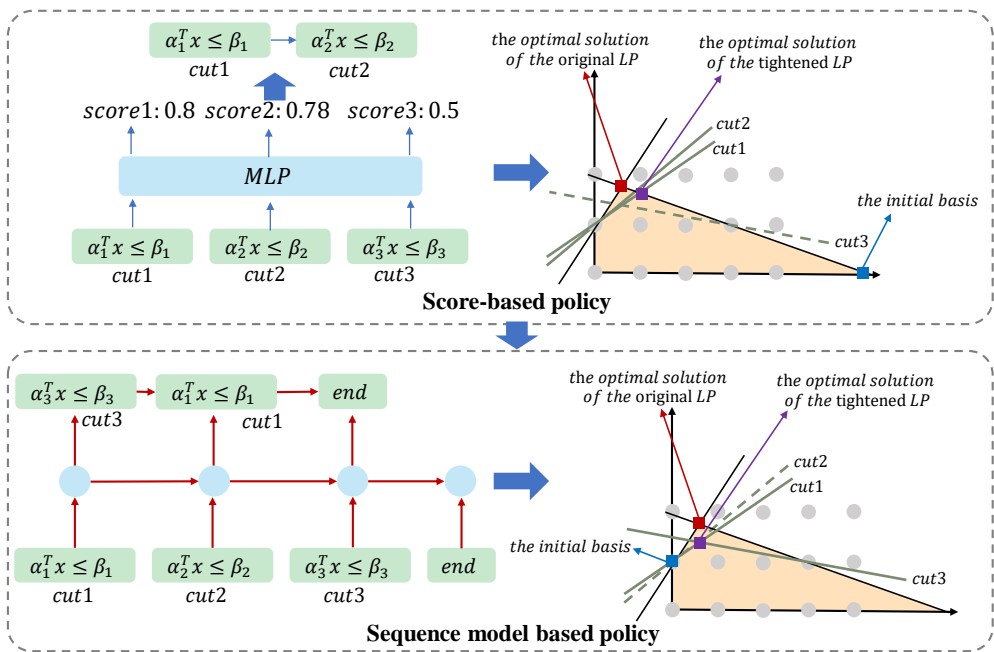

Figure 4: Illustration of selecting cuts using a sequence to sequence model compared to using a scoring function. The sequence model has two main advantages. First, it captures the interaction among cuts by selecting cuts one by one. Consequently, it selects cut3 and cut1 that complement each other nicely, leading to more tightened LP relaxation. Second, it naturally captures the order of selected cuts. Better order of selected cuts may lead to a better initial basis, thus solving the LP relaxation faster (Li et al., 2022) (see Section 3).

the inference time of our proposed HEM given the average number of candidate cuts. The results in Table 6 show that the computational overhead of the HEM is very low.

## D.2 DATASETS USED IN SECTION 3 IN THE MAIN TEXT

In Figure 1a in the main text, we use five challenging datasets, namely D1, D2, D3, D4, and D5, respectively. Specifically, D1 represents MIPLIB mixed supportcase, D2 represents the single instance neos-1456979 from MIPLIB 2017, D3 represents MIPLIB mixed neos, D4 represents Anonymous, and D5 represents the single instance lectsched-5-obj from MIPLIB 2017. In Figure 1b in the main text, we use the dataset MIPLIB mixed neos.

## D.3 LARGE-SCALE REAL-WORLD PRODUCTION PLANNING PROBLEMS

The production planning problem aims to find the optimal production planning for thousands of factories according to the daily order demand. The constraints include the production capacity for each production line in each factory, transportation limit, the order rate, etc. The optimization objective is to minimize the production cost and lead time simultaneously. We split the dataset into training and test sets with $80\%$ and $20\%$ of the instances. The average size of the production planning problems is approximately equal to $3500 \times 5000 = 1.75 \times 10^8$, which are large-scale real-world problems. To promote the machine learning community for MILP, we will release the dataset once the paper is accepted to be published.

## E ILLUSTRATION OF ADVANTAGES OF USING A SEQUENCE MODEL

Figure 4 illustrate two major advantages of using the sequence model to learn cut selection. First, the sequence model takes into account the order of selected cuts by modeling the selected cuts as an output sequence. As shown in Figure 4, the order of cuts determined by the sequence model is better than the score-based method, thus leading to a better initial basis for solving the LP relaxation faster. Second, the sequence model captures the interaction among cuts, as it models the *joint* conditional

probability of the selected cuts given an input sequence of the candidate cuts. As shown in Figure 4, the sequence model selects cuts that complement each other nicely, thus leading to a more tightened LP relaxation and speeding up solving the MILP.

# F  ALGORITHM IMPLEMENTATION AND EXPERIMENTAL SETTINGS

## F.1  DESIGNED CUT FEATURES

Following Huang et al. (2022); Wesselmann & Stuhl (2012); Dey & Molinaro (2018b); Achterberg (2007), we design thirteen cut features for the cut selection task, such as the extent to which a cut is violated by the current LP solution and the proportion of non-zero coefficients of a cut. We

Table 7: The designed cut features of a generated cut $\boldsymbol{\alpha}^T \mathbf{x} \leq \beta$. (Suppose $\mathbf{c}$ denotes the objective coefficient.)

| Feature | Description | Number |
|---|---|---|
| cut coefficients | the mean, max, min, std of cut coefficients | 4 |
| objective coefficients | the mean, max, min, std of the objective coefficients | 4 |
| parallelism | the parallelism between the objective and the cut $\frac{\mathbf{c}^T \boldsymbol{\alpha}}{|\mathbf{c}||\boldsymbol{\alpha}|}$ | 1 |
| efficacy | the Euclidean distance of the cut hyperplane to the current LP solution | 1 |
| support | the proportion of non-zero coefficients of the cut | 1 |
| integral support | the proportion of non-zero coefficients with respect to integer variables of the cut | 1 |
| normalized violation | the violation of the cut to the current LP solution $\max\{0, \frac{\boldsymbol{\alpha}^T \mathbf{x}_{LP}^* - \beta}{|\beta|}\}$ | 1 |

present a detailed description of the designed cut features in Table 7. We emphasize that we do not tune the cut features. Therefore, it is promising to further improve our method by designing better cut features or using graph neural networks to learn better features in future work.

## F.2  IMPLEMENTATION DETAILS OF THE BASELINES

In this part, we present a detailed description of all the baselines used in this paper. We denote a cut by $\boldsymbol{\alpha}^T \mathbf{x} \leq \beta$ and the optimal solution of the current LP relaxation by $\mathbf{x}^*$. Throughout all experiments, we set the ratio of selected cuts as $0.2$ for all score-based rules and learning baselines.

**Random.** Random selects a fixed ratio of the candidate cuts stochastically. The ratio is set as $0.2$ in this paper.

**Normalized Violation (NV).** NV is a score-based rule. It scores each cut based on the normalized violation of the cut to the current LP solution, and selects a fixed ratio of cuts with high scores. The normalized violation is defined by $\max\{0, \frac{\boldsymbol{\alpha}^T \mathbf{x}_{LP}^* - \beta}{|\beta|}\}$. The ratio is set as $0.2$ in this paper.

**Efficacy (Eff).** Eff is a score-based rule. It scores each cut based on the Euclidean distance of the cut hyperplane to the current LP solution, and selects a fixed ratio of cuts with high scores. The ratio is set as $0.2$ in this paper.

**Default.** Default is the default cut selection rule used in SCIP 8.0.0 (Bestuzheva et al., 2021). Please refer to (Achterberg, 2007) for a detailed description of the SCIP's default cut selection rule. Note that Default tackles the two problems: (1) which cuts should be preferred, and (2) how many cuts should be selected, in cut selection by human-designed heuristics. That is, Default selects variable ratios of cuts rather than a fixed ratio.

**Score-based policy (SBP).** Since the state-of-the-art (SOTA) reinforcement learning based method for cut selection (Tang et al., 2020) is designed for the setting that selects the best cut in each round, we implement a slight variant of the SOTA to adapt to our setting that selects a subset of cuts in each round, namely SBP. Specifically, the core idea of SBP is learning a scoring function to measure cut quality as Tang et al. (2020); Huang et al. (2022); Paulus et al. (2022) do. For a fair comparison, SBP uses the same cut features as HEM and we train SBP via reinforcement learning as well. Our implemented SBP is also a slight variant of the method proposed in Huang et al. (2022). We emphasize that experiments in the main text show that our implemented SBP is a strong baseline. Specifically, we implement the scoring function with a multi-layer perceptron that predicts the score of a given cut. That is, the scoring function predicts a cut's score based on the features of the cut. The MLP network contains two hidden layers with 128 units. Moreover, we train the scoring function via evolutionary strategies as (Tang et al., 2020) does. We will also release the code for implementing SBP once the paper is accepted to be published.

---

**Algorithm 2** Pseudo code for training the HEM

---

1: **Initialize** Hierarchical sequence model $\pi_{[\theta_1, \theta_2]}$, MILP instances $\mathcal{D}$, training dataset $\mathcal{D}_{\text{train}}$, batch size $N_b$, training epochs $N_e$, policy learning rate $\alpha$
2: **for** $N_e$ epochs **do**
3:     Empty the training dataset $\mathcal{D}_{\text{train}}$
4:     **for** $N_b$ steps **do**
5:         Randomly sample a MILP $s_0$ from $\mathcal{D}$
6:         Take action $k$ and $a_k$ at state $s_0$ with the policy $\pi$
7:         Receive reward $r$ and add $(s_0, k, a_k, r)$ to $\mathcal{D}_{\text{train}}$
8:     **end for**
9:     Compute hierarchical policy gradient using $\mathcal{D}_{\text{train}}$ as in proposition 1
10:    Update the parameters, $\theta_1 = \theta_1 + \alpha \nabla_{\theta_1} J([\theta_1, \theta_2])$, $\theta_2 = \theta_2 + \alpha \nabla_{\theta_2} J([\theta_1, \theta_2])$
11: **end for**

---

### F.3 IMPLEMENTATION DETAILS AND HYPERPARAMETERS

#### F.3.1 HARDWARE SPECIFICATION

Throughout all experiments, we use a single machine that contains eight GPU devices (NVidia GeForce GTX 3090 Ti) and two Intel Gold 6246R CPUs.

#### F.3.2 SOLVER SETUP

For reproducibility, we emphasize that all results in the main text are obtained by averaging results over the SCIP random seeds $\{1, 2, 3\}$.

#### F.3.3 REWARD FUNCTION

On the easy datasets, we set the reward as the negative solving time. On the medium and hard datasets, we set the reward as the negative primal-dual gap integral within a time limit of 300 seconds.

For the real-world production planning problems, we set the reward as the negative primal-dual gap integral within a time limit of 600 seconds or the negative dual bound improvement. The results reported in the main text are achieved by HEM with the negative dual bound improvement reward. We provide the performance of HEM that uses the negative primal-dual gap integral in Table 8. The results still show that HEM significantly outperforms all the baselines in terms of the Time and PD integral.

Table 8: Evaluation on real-world production planning problems with rewards being the negative PD integral. The results show that HEM still significantly outperforms all the baselines.

| | Production planning | | | |
|---|---|---|---|---|
| Method | Time (s) | Improvement (Time, %) | PD integral | Improvement (PD integral, %) |
| NoCuts | 278.79 | NA | 17866.01 | NA |
| Default | 296.12 | -6.22 | 17703.39 | 0.91 |
| Random | 280.18 | -0.50 | 18120.21 | -1.42 |
| NV | 259.48 | 6.93 | 17295.18 | 3.20 |
| Eff | 263.60 | 5.45 | 16636.52 | 6.88 |
| SBP | 276.61 | 0.78 | 16952.85 | 5.11 |
| HEM (Ours) | **251.64** | **9.74** | **16533.05** | **7.46** |

We emphasize that we can set the reward according to our objective in real-world problems. For example, suppose we aim to minimize the primal-dual gap within a time limit, then we can set the reward as the primal-dual gap within the time limit.

#### F.3.4 POLICY NETWORK ARCHITECTURE

The higher-level model contains an LSTM encoder and an MLP. The LSTM network encodes variable-sized inputs into hidden vectors with dimension 128. The MLP network contains two hidden layers with 128 units. The lower-level model is essentially a pointer network. We keep the hyperparameters of the pointer network as that used in (Bello* et al., 2017).

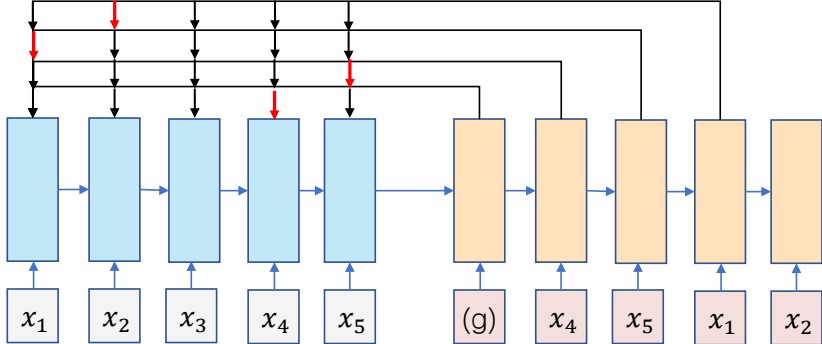

Figure 5: Illustration of the pointer network architecture introduced by Vinyals et al. (2015).

### F.3.5 OPTIMIZATION

Throughout all experiments, we apply Adam optimizer with learning rate $\alpha_1 = 1 \times 10^{-4}$ to optimize the lower-level model, and learning rate $\alpha_2 = 5 \times 10^{-4}$ to optimize the higher-level model. For each epoch, we collect 32 samples for training, and we set the total epochs as 100. It is surprising that learning a good cut selection policy does not need too much data as shown in Tang et al. (2020). For training stability, we delay the higher-level policy update. This creates a two-timescale algorithm, as often required for convergence in the linear setting (Fujimoto et al., 2018; Konda & Tsitsiklis, 2003). We set the delay update freq as two. That is, we first train the lower-level policy twice, then train the higher-level policy once. Additionally, the results in Appendix G.3.4 show that the convergence performance of HEM is insensitive to the hyperparameter delay update freq.

### F.3.6 THE TRAINING ALGORITHM

We provide the procedure of the training algorithm of HEM in Algorithm 2.

## F.4 MORE DETAILS OF HEM

### F.4.1 DETAILS OF THE POINTER NETWORK

The pointer network is first introduced by (Vinyals et al., 2015) for directly finding solutions of specific combinatorial optimization problems, such as the Travelling Salesman Problems. The pointer network architecture is illustrated in Figure 5. The pointer network consists of a Long Short-Term Memory encoder, a Long Short-Term Memory decoder, and an attention that is used as a pointer to select a member of the input sequence as the output (Vinyals et al., 2015). Specifically, we implement the pointer network following Bello* et al. (2017). Please refer to (Bello* et al., 2017) for implementation details of the pointer network.

The major difference between our used pointer network and the pointer network used in (Bello* et al., 2017) is that we use the pointer network to select ordered subsets of input sequences, but (Bello* et al., 2017) use the pointer network to output permutations of input sequences.

### F.4.2 DIFFERENCE BETWEEN HEM-RATIO AND HEM W/O H

**Details of HEM w/o H** To implement HEM w/o H, we augment each input sequence with an end token, i.e., a thirteen-dimensional tensor with values all being one. The end token is at the end position of the input sequence. Once the decoder of HEM w/o H outputs the end token, then the decoding ends. That is, HEM w/o H can implicitly predict the number of cuts that should be selected by predicting whether to decode the end token at the current decoding step.

The policy network of HEM-ratio and HEM w/o H are both essentially a pointer network (Vinyals et al., 2015), a variant of the sequence model. We present the major difference between HEM-ratio and HEM w/o H in the following. HEM w/o H predicts an end token as used in language tasks (Sutskever et al., 2014; Vaswani et al., 2017) to determine the number of cuts that should be selected implicitly. In contrast, HEM-ratio always selects a fixed ratio of cuts, i.e, it always ends decoding

Table 9: Policy evaluation on easy, medium, and hard datasets in terms of the total number of nodes and the primal-dual gap. The best performance are marked in bold.

| | Set Covering | | Maximum Independent Set | | Multiple Knapsack | |
|---|---|---|---|---|---|---|
| Method | Nodes | PD gap | Nodes | PD gap | Nodes | PD gap |
| NoCuts | 189.44 (423.68) | 0.00 (0) | 2170.66 (4054.09) | 0.00 (0) | 16945.58 (41242.04) | 0.00 (0.000128) |
| Default | 116.77 (420.09) | 0.00 (0) | 588.23 (1916.99) | 0.00 (0) | 16949.88 (41297.50) | 0.00 (0.000128) |
| Random | 95.70 (285.05) | 0.00 (0) | 1416.96 (3820.01) | 0.00 (0) | 21463.16 (59411.07) | 0.00 (0.000361) |
| NV | 199.70 (436.25) | 0.00 (0) | 1618.42 (3089.12) | 0.00 (0) | 20673.32 (62526.13) | 0.00 (0.00022) |
| Eff | 194.17 (439.35) | 0.00 (0) | 1575.88 (2742.66) | 0.00 (0) | 14909.93 (28575.12) | 0.00 (0) |
| SBP | 1.16 (2.04) | 0.00 (0) | 698.41 (2869.65) | 0.00 (0) | 13537.59 (22527.45) | 0.00 (0) |
| HEM (Ours) | **1.11 (1.46)** | **0.00 (0)** | **311.64 (1309.94)** | **0.00 (0)** | **10463.85 (18491.25)** | **0.00 (0)** |

| | MIK | | CORLAT | | Load Balancing | |
|---|---|---|---|---|---|---|
| Method | Nodes | PD gap | Nodes | PD gap | Nodes | PD gap |
| NoCuts | 395290.17 (120526.21) | 0.09 (0.018) | 97098.72 (119287.56) | 2.000000e+18 (1.4E+19) | 108.20 (56.28) | 0.94 (0.12) |
| Default | 224746.60 (179450.80) | 0.02 (0.033) | 70215.01 (113515.62) | 2.666667e+18 (1.61E+19) | **65.22 (55.34)** | 0.43 (0.073) |
| Random | 403618.10 (115802.29) | 0.07 (0.035) | 74149.90 (109043.28) | 2.000000e+18 (1.4E+19) | 107.05 (68.05) | 0.79 (0.133) |
| NV | 397025.47 (106676.73) | 0.08 (0.028) | 77943.73 (111355.03) | 4.666667e+18 (2.11E+19) | 81.91 (38.55) | 0.82 (0.11) |
| Eff | 406832.83 (113363.27) | 0.07 (0.033) | 95956.53 (120858.92) | 5.333333e+18 (2.25E+19) | 86.22 (45.35) | 0.82 (0.11) |
| SBP | 417070.37 (133702.50) | 0.07 (0.030) | 62297.99 (109664.21) | 6.666667e+17 (8.14E+18) | 138.37 (53.79) | 0.65 (0.06) |
| HEM(Ours) | **220547.93 (172537.94)** | **0.02 (0.028)** | **51929.71 (100973.74)** | **0.02 (0.051)** | 74.47 (60.98) | **0.42 (0.072)** |

| | Anonymous | | MIPLIB mixed neos | | MIPLIB mixed supportcase | |
|---|---|---|---|---|---|---|
| Method | Nodes | PD gap | Nodes | PD gap | Nodes | PD gap |
| NoCuts | 20110.58 (14167.51) | 8.333333e+18 (2.76E+19) | 169348.83 (117745.23) | 2.500000e+19 (4.33E+19) | 5971.00 (8679.98) | 9.19 (19.29) |
| Default | 20025.78 (13056.50) | 8.333333e+18 (2.76E+19) | 161366.25 (107049.96) | 2.500000e+19 (4.33E+19) | **4055.04 (7404.61)** | 11.73 (31.52) |
| Random | 19824.48 (12366.55) | 1.000000e+19 (3E+19) | **143930.17 (95296.41)** | 2.500000e+19 (4.33E+19) | 4656.13 (7745.46) | 6.05 (14.69) |
| NV | **19313.33 (12391.94)** | 8.333333e+18 (2.76E+19) | 150046.50 (94801.62) | 2.500000e+19 (4.33E+19) | 4986.21 (9024.19) | 2.78 (10.08) |
| Eff | 19526.23 (12116.80) | 6.666667e+18 (2.49E+19) | 144128.83 (93404.76) | 2.500000e+19 (4.33E+19) | 4450.88 (7845.48) | 5.58 (15.40) |
| SBP | 19351.67 (12337.81) | 6.666667e+18 (2.49E+19) | 177736.25 (122005.81) | 2.500000e+19 (4.33E+19) | 5618.08 (9765.43) | 0.14 (0.26) |
| HEM(Ours) | 20191.28 (13219.21) | **1.666667e+18 (1.28E+19)** | 177735.83 (129020.42) | **2.500000e+19 (4.33E+19)** | 4844.88 (9996.16) | **0.14 (0.25)** |

at a pre-determined position. Therefore, HEM w/o H can learn the number of cuts that should be selected, but HEM-ratio cannot.

### F.4.3 MORE DISCUSSION OF HEM

In this part, we provide details of some more advantages of HEM. (1) Inspired by hierarchical reinforcement learning (Sutton et al., 1999b; Nachum et al., 2018), HEM leverages the hierarchical structure of the cut selection task, which is important for efficient exploration in complex decision-making tasks. (2) Previous methods (Tang et al., 2020; Huang et al., 2022) usually train cut selection policies via black-box optimization methods such as evolution strategies (Salimans et al., 2017). In contrast, HEM is differentiable and we train the HEM via gradient-based algorithms, which is more sample efficient than black-box optimization methods (Sutton & Barto, 2018; Schulman et al., 2015). Although we can offline generate training samples as much as possible using a MILP solver, high sample efficiency is significant as generating samples can be extremely time-consuming in practice.

## G MORE RESULTS

### G.1 MORE MOTIVATING RESULTS

**Ratio matters.** To evaluate the effect of the ratio of selected cuts on solving MILPs, we focus on the Normalized Violation (NV) cut selection method with different ratios of selected cuts. (1) We first evaluate the NV methods that select $5\%, 10\%, 20\%, 30\%, 40\%, 50\%, 60\%, 70\%$, and $80\%$ of candidate cuts, respectively, on four datasets. The results in Figure 6a show that the NV achieves better solver performance with larger ratios on CORLAT and MIPLIB mixed neos. The results demonstrate that the ratio that leads to better solver performance is variable over different datasets, which implies that learning dataset-dependent ratios is important. (2) We then evaluate the NV methods that select $5\%, 10\%, 20\%, 30\%, 40\%, 50\%, 60\%, 70\%$, and $80\%$ of candidate cuts, respectively, on four instances from the Anonymous dataset. The results in Figure 6b show that NV achieves better solver performance with larger ratios on Anonymous 121 and Anonymous 131. The results demonstrate that the ratio that leads to better solver performance is variable over different instances from the same dataset, which implies that learning instance-dependent ratios is important as well.

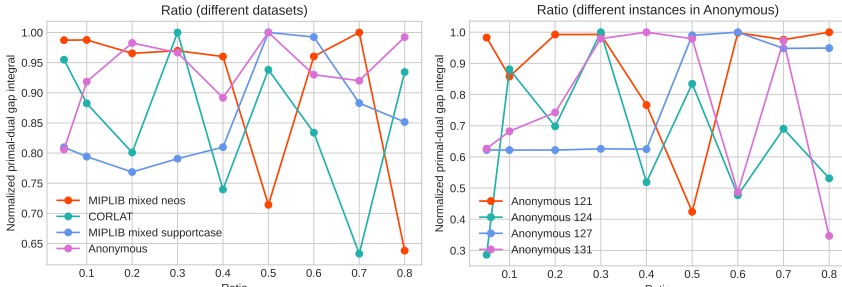

(a) NV with different ratios on four datasets. (b) NV with different ratios on Anonymous.

Figure 6: We use the Normalized Violation (NV) rule (Huang et al., 2022). The results in (c) show that NV with different given ratios achieve variable normalized PD integral on four datasets. The results in (d) show that NV with different ratios achieve variable normalized PD integral on different instances from the same dataset.

## G.2 MORE RESULTS OF MAIN EVALUATION

In this section, we provide more results of the main evaluation. The results in Table 9 show the performance of HEM and the baselines in terms of the total number of nodes (Nodes) and the primal-dual gap (PD gap). (1) **Easy datasets.** On the easy datasets, HEM and most baselines find the optimal solution within the time limit, as the PD gap converges to zero. Additionally, HEM significantly outperforms all the baselines in terms of the Nodes on the easy datasets. (2) **Medium and hard datasets.** In terms of the PD gap, HEM outperforms all the baselines on medium and hard datasets, especially on CORLAT. However, the Nodes metric cannot well distinguish the performance of different cut selection methods on the medium and hard datasets for the following two reasons. First, the solving time of the LP relaxation on each node is different, and thus the Nodes cannot directly determine the solving time (Huang et al., 2022). Second, on those unsolved instances within the time limit, the Nodes metric is not a proper metric, as the Nodes cannot evaluate the quality of the solving process.

## G.3 MORE RESULTS OF ABLATION STUDY

In this section, we provide more results of ablation studies in the main text.

### G.3.1 IN-DEPTH ANALYSIS OF HEM-RATIO AND SBP

We provide possible reasons for HEM-ratio performing poorer than SBP on several challenging MILP problem benchmarks. Fundamentally, HEM-ratio formulates the cut selection task as a sequence modeling problem, which has two main advantages over SBP. That is, the sequence model can not only capture the underlying order information, but also capture the interaction among cuts. However, training a sequence model is more difficult than training a scoring function, as the sequence model aims to learn a much more complex task. Specifically, the scoring function aims to learn to score each cut, while the sequence model aims to model the joint probability of the selected cuts. The latter is a more challenging learning task. Moreover, we follow the reinforcement learning paradigm instead of supervised learning to train the model, making the training process more unstable. Therefore, the sequence model may suffer from inefficient exploration and be trapped to a local optimum. As a result, HEM-ratio can perform poorer than SBP, especially on challenging MILP problem benchmarks.

### G.3.2 CONTRIBUTION OF EACH COMPONENT

To understand the contribution of each component of HEM, we provide more results of HEM and HEM without the higher-level model on Set Covering, Multiple Knapsack, MIK, Load Balancing, Anonymous, and MIPLIB mixed supportcase. The results in Table 10 show that HEM outperforms HEM w/o H in terms of the solving time, the primal-dual gap, and the primal-dual gap integral on several challenging datasets, demonstrating the importance of our proposed higher-level model. Moreover, HEM w/o H significantly outperforms SBP in terms of the solving time, the primal-dual

Table 10: Comparsion between HEM, HEM without the higher-level model on more datasets.

| | Set Covering | | | | Multiple Knapsack | | | |
|---|---|---|---|---|---|---|---|---|
| Method | Time (s) | Nodes | PD gap | PD integral | Time (s) | Nodes | PD gap | PD integral |
| NoCuts | 6.31 (4.61) | 189.44 (423.68) | 0.00 (0) | 56.99 (38.89) | 9.88 (22.24) | 16945.58 (41242.04) | 0.00 (0.000128) | 16.41 (14.16) |
| Default | 4.41 (5.12) | 116.77 (420.08) | 0.00 (0) | 55.63 (42.21) | 9.90 (22.24) | 16949.87 (41297.49) | 0.00 (0.000128) | 16.46 (14.25) |
| SBP | 1.91 (0.36) | 1.16 (2.04) | 0.00 (0) | 38.96 (8.66) | 7.74 (12.36) | 13537.58 (22527.45) | 0.00 (0) | 16.45 (16.62) |
| HEM w/o H | **1.84 (0.31)** | 1.18 (2.03) | 0.00 (0) | **37.69 (8.33)** | 7.36 (12.90) | 12418.85 (23256.47) | 0.00 (0) | 14.01 (10.94) |
| HEM (Ours) | 1.85 (0.31) | **1.09 (1.46)** | **0.00 (0)** | 37.92 (8.46) | **6.13 (9.61)** | **10463.84 (18491.25)** | **0.00 (0)** | **13.63 (9.63)** |

| | MIK | | | | Load Balancing | | | |
|---|---|---|---|---|---|---|---|---|
| Method | Time (s) | Nodes | PD gap | PD integral | Time (s) | Nodes | PD gap | PD integral |
| NoCuts | 300.01 (0.009) | 395290.16 (120526.21) | 0.09 (0.018) | 2355.87 (996.08) | 300.00 (0.12) | 108.20 (56.28) | 0.94 (0.12) | 14853.77 (951.42) |
| Default | 179.62 (122.36) | 224746.60 (179450.80) | 0.02 (0.033) | 844.40 (924.30) | 300.00 (0.06) | 65.22 (55.34) | 0.43 (0.07) | 9589.19 (1012.95) |
| SBP | 286.07 (41.81) | 417070.36 (133702.49) | 0.07 (0.0295) | 2053.30 (740.11) | 300.00 (0.10) | 138.37 (53.79) | 0.65 (0.06) | 12535.30 (741.43) |
| HEM w/o H | 218.87 (115.97) | 297058.97 (189410.43) | 0.04 (0.039) | 1321.63 (1165.23) | 300.03 (0.04) | 70.93 (60.36) | **0.42 (0.07)** | 9475.16 (1005.81) |
| HEM (Ours) | **176.12 (125.18)** | **220547.93 (172537.94)** | **0.02 (0.028)** | **785.04 (790.38)** | **300.00 (0.04)** | 74.47 (60.98) | 0.42 (0.07) | 9496.42 (1018.35) |

| | Anonymous | | | | MIPLIB mixed supportcase | | | |
|---|---|---|---|---|---|---|---|---|
| Method | Time (s) | Nodes | PD gap | PD integral | Time (s) | Nodes | PD gap | PD integral |
| NoCuts | 246.22 (94.90) | 20110.58 (14167.51) | 8.333333e+18 (2.76E+19) | 18297.30 (9769.42) | 170.00 (131.60) | 5971.00 (8679.98) | 9.19 (19.29) | 9927.96 (11334.07) |
| Default | 244.02 (97.72) | 20025.78 (13056.50) | 8.333333e+18 (2.76E+19) | 17407.01 (9736.19) | 164.61 (135.82) | 4055.04 (7404.61) | 11.73 (31.52) | 9672.34 (10668.24) |
| SBP | 245.71 (92.46) | **19351.66 (12337.81)** | 6.666667e+18 (2.49E+19) | 18188.63 (9651.85) | 165.61 (135.25) | 5618.08 (9765.43) | 0.14 (0.26) | 7408.65 (7903.47) |
| HEM w/o H | 251.00 (88.45) | 21192.40 (16436.32) | 5.000000e+18 (2.18E+19) | 17226.84 (9553.90) | **156.96 (133.35)** | 3779.83 (7465.75) | 0.56 (1.43) | 7709.19 (8655.48) |
| HEM (Ours) | **241.68 (97.23)** | 20191.28 (13219.21) | **1.666667e+18 (1.28E+19)** | **16077.15 (9108.21)** | 162.96 (138.21) | 4844.87 (9996.15) | **0.14 (0.25)** | **6874.80 (6729.97)** |

Table 11: Comparsion between HEM, HEM-ratio, and HEM-ratio-order on more datasets.

| | Set Covering | | | | Multiple Knapsack | | | |
|---|---|---|---|---|---|---|---|---|
| Method | Time (s) | Nodes | PD gap | PD integral | Time (s) | Nodes | PD gap | PD integral |
| NoCuts | 6.31 (4.61) | 189.44 (423.68) | 0.00 (0) | 56.99 (38.89) | 9.88 (22.24) | 16945.58 (41242.04) | 0.00 (0.000128) | 16.41 (14.16) |
| Default | 4.41 (5.12) | 116.77 (420.08) | 0.00 (0) | 55.63 (42.21) | 9.90 (22.24) | 16949.87 (41297.49) | 0.00 (0.000128) | 16.46 (14.25) |
| SBP | 1.91 (0.36) | 1.16 (2.04) | 0.00 (0) | 38.96 (8.66) | 7.74 (12.36) | 13537.58 (22527.45) | 0.00 (0) | 16.45 (16.62) |
| HEM-ratio-order | 2.11 (0.38) | 1.11 (1.29) | 0.00 (0) | 42.01 (9.88) | 10.46 (29.74) | 18434.20 (59254.10) | 0.00 (0.000128) | 16.92 (18.19) |
| HEM-ratio | 2.10 (0.40) | 1.11 (1.18) | 0.00 (0) | 41.95 (9.82) | 7.63 (12.64) | 13307.73 (24339.3) | 0.00 (0) | 16.19 (15.16) |
| HEM (Ours) | **1.85 (0.31)** | **1.09 (1.46)** | **0.00 (0)** | **37.92 (8.46)** | **6.13 (9.61)** | **10463.84 (18491.25)** | **0.00 (0)** | **13.63 (9.63)** |

| | MIK | | | | Load Balancing | | | |
|---|---|---|---|---|---|---|---|---|
| Method | Time (s) | Nodes | PD gap | PD integral | Time (s) | Nodes | PD gap | PD integral |
| NoCuts | 300.01 (0.009) | 395290.16 (120526.21) | 0.09 (0.018) | 2355.87 (996.08) | 300.00 (0.12) | 108.20 (56.28) | 0.94 (0.12) | 14853.77 (951.42) |
| Default | 179.62 (122.36) | 224746.60 (179450.80) | 0.02 (0.033) | 844.40 (924.30) | 300.00 (0.06) | **65.22 (55.34)** | 0.43 (0.07) | 9589.19 (1012.95) |
| SBP | 286.07 (41.81) | 417070.36 (133702.49) | 0.07 (0.0295) | 2053.30 (740.11) | 300.00 (0.10) | 138.37 (53.79) | 0.65 (0.06) | 12535.30 (741.43) |
| HEM-ratio-order | 282.92 (42.07) | 417397.10 (132077.3) | 0.06 (0.034) | 2072.69 (849.16) | 300.11 (0.11) | 145.18 (56.52) | 0.65 (0.061) | 12368.12 (726.77) |
| HEM-ratio | 283.75 (39.87) | 401540.97 (131295.8) | 0.07 (0.034) | 1869.66 (978.85) | 300.10 (0.098) | 148.92 (59.12) | 0.65 (0.058) | 12410.84 (715.44) |
| HEM (Ours) | **176.12 (125.18)** | **220547.93 (172537.94)** | **0.02 (0.028)** | **785.04 (790.38)** | **300.00 (0.04)** | 74.47 (60.98) | **0.42 (0.07)** | **9496.42 (1018.35)** |

| | Anonymous | | | | MIPLIB mixed supportcase | | | |
|---|---|---|---|---|---|---|---|---|
| Method | Time (s) | Nodes | PD gap | PD integral | Time (s) | Nodes | PD gap | PD integral |
| NoCuts | 246.22 (94.90) | 20110.58 (14167.51) | 8.33e+18 (2.76E+19) | 18297.30 (9769.42) | 170.00 (131.60) | 5971.00 (8679.98) | 9.19 (19.29) | 9927.96 (11334.07) |
| Default | 244.02 (97.72) | 20025.78 (13056.50) | 8.33e+18 (2.76E+19) | 17407.01 (9736.19) | 164.61 (135.82) | **4055.04 (7404.61)** | 11.73 (31.52) | 9672.34 (10668.24) |
| SBP | 245.71 (92.46) | **19351.66 (12337.81)** | 6.67e+18 (2.49E+19) | 18188.63 (9651.85) | 165.61 (135.25) | 5618.08 (9765.43) | 0.14 (0.26) | 7408.65 (7903.47) |
| HEM-ratio-order | 245.45 (94.99) | 20495.80 (12472.44) | 5.00e+18 (2.18E+19) | 16496.96 (9282.15) | 169.45 (132.55) | 6252.63 (9827.98) | 4.25 (15.97) | 9226.95 (9995.94) |
| HEM-ratio | 245.17 (95.21) | 20942.07 (13379.46) | 5.00e+18 (2.18E+19) | 16148.82 (9247.48) | 163.03 (137.16) | 5551.29 (10708.46) | 7.54 (21.85) | 9979.35 (11048.11) |
| HEM (Ours) | **241.68 (97.23)** | 20191.28 (13219.21) | **1.67e+18 (1.28E+19)** | **16077.15 (9108.21)** | 162.96 (138.21) | 4844.87 (9996.15) | **0.14 (0.25)** | **6874.80 (6729.97)** |

gap, and the primal-dual gap integral on several challenging datasets, demonstrating the significance of our proposed lower-level model.

### G.3.3 THE IMPORTANCE OF TACKLING **P1-P3** IN CUT SELECTION

To understand the importance of tackling **P1-P3** in cut selection, we provide more results of HEM, HEM-ratio, and HEM-ratio-order on Set Covering, Multiple Knapsack, MIK, Load Balancing, Anonymous, and MIPLIB mixed supportcase. Here we refresh what HEM, HEM-ratio, HEM-ratio-order mean. (1) **HEM**. HEM tackles **P1-P3** in cut selection simultaneously. (2) **HEM-ratio**. In order not to learn how many cuts should be selected, we remove the higher-level model of HEM and force the lower-level model to select a fixed ratio of cuts. We denote it by HEM-ratio. Note that HEM-ratio is different from HEM w/o H (see Appendix F). HEM-ratio tackles **P1** and **P3** in cut selection. (3) **HEM-ratio-order.** To further mute the effect of the order of selected cuts, we reorder the selected cuts given by HEM-ratio with the original index of the generated cuts, which we denote by HEM-ratio-order. HEM-ratio-order mainly tackles **P1**.

The results in Table 11 suggest the following. HEM-ratio-order outperforms Default and NoCuts on several datasets, demonstrating that tackling **P1** by data-driven methods is crucial. HEM significantly outperforms HEM-ratio in terms of the primal-dual gap integral, demonstrating the significance of tackling **P2**. HEM-ratio outperforms HEM-ratio-order on several datasets, which demonstrates the importance of tackling **P3**. Moreover, HEM-ratio performs better than SBP in terms of

Table 12: Sensitivity analysis of HEM to the hyperparameter dealy update freq $d$.

| | Maximum Independent Set | | | Corlat | | | MIPLIBS Mixed (neos) | | |
| Method | Time(s) | PD integral | Improvement (PD integral, %) | Time(s) | PD integral | Improvement (PD integral, %) | Time(s) | PD integral | Improvement (PD integral, %) |
|---|---|---|---|---|---|---|---|---|---|
| NoCuts | 8.78 (6.66) | 71.31 (51.74) | NA | 103.30 (128.14) | 2818.40 (5908.31) | NA | 253.65 (80.29) | 14652.29 (12523.37) | NA |
| HEM (delay=1) | 1.79 (3.65) | 16.56 (26.6) | 76.78 | 57.39 (111.76) | 1260.83 (3518.07) | 55.26 | 254.48 (78.84) | 9273.01 (12031.44) | 36.71 |
| HEM (delay=2) | 1.76 (3.69) | 16.01 (26.21) | 77.55 | 58.31 (110.51) | 1079.99 (2653.14) | 61.68 | 248.66 (89.46) | 8678.76 (12337.00) | 40.77 |
| HEM (delay=3) | 1.80 (3.82) | 16.91 (28.49) | 76.29 | 58.50 (108.88) | 1003.52 (2264.91) | 64.39 | 247.62 (90.73) | 8408.16 (12467.02) | 42.62 |
| HEM (delay=4) | 1.89 (3.90) | 17.34 (28.46) | 75.68 | 69.60 (115.42) | 1309.33 (3336.59) | 53.54 | 246.93 (91.93) | 8368.46 (12489.68) | 42.89 |

Table 13: Evaluate the generalization ability of HEM on Set Covering.

| | Set Covering (2×) | | | Set Covering (4×) | | |
| Method | Time(s) ↓ | Improvement ↑ (time, %) | PD integral ↓ | Time(s) ↓ | Improvement ↑ (time, %) | PD integral ↓ |
|---|---|---|---|---|---|---|
| NoCuts | 82.69 (78.27) | NA | 609.43 (524.92) | 284.44 (48.70) | NA | 3215.34 (1019.47) |
| Default | 61.01 (78.12) | 26.22 | 494.63 (545.76) | 149.69 ( 141.92) | 47.37 | 1776.22 (1651.15) |
| Random | 64.44 (73.98) | 22.07 | 520.84 (489.52) | 208.12 (131.52) | 26.53 | 2528.36 (1678.66) |
| NV | 92.05 (80.11) | -11.32 | 725.53 (541.68) | 286.10 (45.47) | -0.58 | 3422.46 (1024.19) |
| Eff | 92.32 (79.33) | -11.64 | 733.72 (538.60) | 286.20 (45.04) | -0.62 | 3437.06 (1043.44) |
| SBP | 3.52 (1.36) | 95.74 | 92.89 (25.83) | 7.62 (6.46) | 97.32 | 256.79 (145.92) |
| HEM (Ours) | **3.33 (0.47)** | **95.97** | **89.24 (14.26)** | **7.40 (5.03)** | **97.40** | **250.83 (131.43)** |

the solving time and the primal-dual gap integral on all six datasets except Set Covering and MI-PLIB mixed supportcase, which shows the superiority of formulating the cut selection as a sequence to sequence learning problem over formulating it as a scoring task. However, HEM-ratio and HEM-ratio-order perform a little worse than SBP on Set Covering and MIPLIB mixed supportcase. A possible reason is that it is more difficult to train a sequence model than to train a multi-layer perceptron and thus the sequence model may suffer from inefficient exploration and be trapped to the local optimum. Please refer to Appendix G.3.1 for a detailed analysis.

### G.3.4 SENSITIVITY ANALYSIS

Additionally, we perform ablation studies to test the sensitivity of HEM to the hyperparameter delay update freq $d$. The results in Table 12 show that there is a wide range of $d$ for HEM to achieve comparable performance on Maximum Independent Set, CORLAT, and MIPLIB mixed neos. Moreover, we emphasize that we do not tune the hyperparameter $d$. As the results shown in Table 12, $d = 3$ and $d = 4$ performs the best on CORLAT and MIPLIB mixed neos, respectively. However, we simply set $d = 2$ throughout all experiments in the main text.

### G.4 MORE RESULTS OF GENERALIZATION

Here we provide more results of the generalization experiments on Set Covering. On Set Covering, we test HEM on two times and four times larger instances than those seen during training. The results in Table 13 show that HEM generalizes well to instances that are significantly larger than seen during training. In particular, HEM achieves at least $70\%$ improvement in terms of the Time compared to all the rule-based baselines. Moreover, SBP also generalizes well to large instances, demonstrating that SBP is a strong baseline.

### G.5 MORE VISUALIZATION RESULTS

In this part, we provide more visualization results on Set Covering and MIK. On MIK, we visualize the cuts selected by HEM-ratio and SBP on a randomly sampled instance. We perform principal component analysis (Mohri et al., 2018) on selected cuts to reduce the cut features to two-dimensional space. Colored points illustrate reduced cut features. To visualize the diversity of selected cuts, we use dashed lines to connect the points with the smallest and largest x,y coordinates. The results in Figure 7 still show that HEM-ratio selects much more diverse cuts than SBP on MIK. However, HEM-ratio performs poorer than SBP on Set Covering (see Appendix G.3.1 for a detailed analysis). Therefore, we visualize the cuts selected by HEM and SBP on a randomly sampled instance from Set Covering. Although HEM learns the number of cuts that should be selected, we find that HEM selects much fewer cuts than SBP. Specifically, HEM selects 25 cuts, while SBP selects 158 cuts. Interestingly, the results in Figure 7 show that SBP selects 158 similar cuts with high scores, while HEM selects much more diverse cuts than SBP. The results show that HEM tends to select cuts that complement each other nicely.

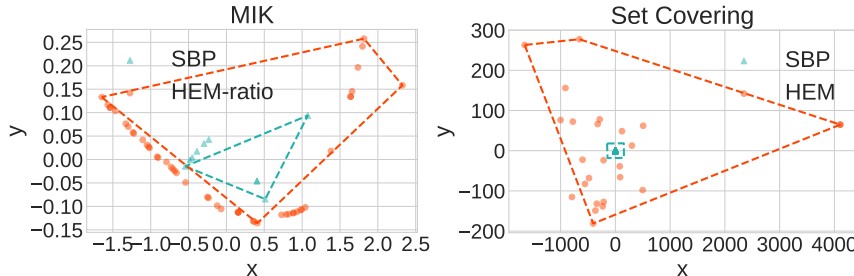

Figure 7: We perform principal component analysis on cuts selected by HEM-ratio/HEM and SBP. Each colored point illustrates a reduced cut feature. To visualize the diversity of selected cuts, we use dashed lines to connect the points with the smallest and largest x,y coordinates.

Table 14: Policy evaluation on MIPLIB mixed neos and MIPLIB mixed supportcase with a time limit of *three hours*.

| | MIPLIB mixed neos | | | | | MIPLIB mixed supportcase | | | | |
|---|---|---|---|---|---|---|---|---|---|---|
| Method | Time (s) | Nodes | PD gap | PD integral | Improvement (PD integral, %) | Time (s) | Nodes | PD gap | PD integral | Improvement (PD integral, %) |
| NoCuts | 8,126.15 (4631.32) | 5,058,282.42 (5067358) | 0.97 (1.14) | 369,180.71 (368274) | NA | 3,616.09 (4703.68) | 258,543.71 (692821) | 0.00 (0.0096) | 37,858.65 (59578.6) | NA |
| Default | 8,129.27 (4625.92) | 4,800,372.67 (4815222) | 0.94 (1.07) | 392,324.09 (393266) | -6.27 | 3,997.69 (4720.2) | 188,844.71 (599665) | 0.01 (0.012) | 56,375.64 (106694) | -48.91 |
| Random | 8,125.44 (4632.53) | 5,107,627.25 (5114152) | 0.95 (1.09) | 395,694.74 (393445) | -7.18 | 2,605.78 (3993.84) | **8,458.96 (14785.9)** | 0.01 (0.029) | 44,222.39 (70937.6) | -16.81 |
| NV | 8,133.64 (4618.37) | 4,930,498.92 (4930971) | 0.91 (1.03) | 326,315.04 (391163) | 11.61 | **2,561.10 (3980.44)** | 34,851.08 (142069) | 0.00 (0.0095) | 27,908.00 (37742.2) | 26.28 |
| Eff | 8,130.78 (4623.31) | 4,788,568.75 (4804443) | 0.94 (1.07) | 395,640.44 (393572) | -7.17 | 3,105.98 (4308.57) | 182,187.63 (543167) | 0.01 (0.028) | 39,932.37 (58307.3) | -5.48 |
| SBP | 8,130.63 (4626.62) | 4,763,378.00 (4829037) | 0.91 (1.03) | 388,564.35 (398176.3) | -5.25 | 3,014.63 (4082.19) | 146,180.96 (543662.45) | 0.00 (0.0084) | 24,447.37 (55984.9) | 35.42 |
| HEM(Ours) | **8,124.32 (4634.54)** | **4,748,599.50 (4816327)** | **0.78 (0.82)** | **194,557.87 (345035)** | **47.30** | 3,465.48 (4485.11) | 52,750.08 (140410) | **0.00 (0.0087)** | **17,885.98 (24045.9)** | **52.76** |

## G.6 EVALUATION WITH A TIME LIMIT OF THREE HOURS

In this section, we aim to evaluate whether HEM can generalize well to solving problems within a much longer time limit. Specifically, we evaluate HEM on two extremely challenging MIPLIB datasets within a time limit of *three hours*. Note that we still train HEM with a time limit of 300 seconds, while we test HEM with a time limit of three hours. The results in Table 14 show that HEM still significantly outperforms all the baselines, especially in terms of the primal-dual gap integral on MIPLIB mixed neos and MIPLIB mixed supportcase. In terms of the primal-dual gap, HEM also outperforms the baselines. Moreover, HEM performs better than baselines in terms of the solving time on MIPLIB mixed neos, but HEM performs poorly in terms of the solving time on MIPLIB mixed supportcase. Interestingly, the primal-dual gap integral is not always consistent with the solving time. We emphasize that we train with the negative primal-dual gap integral reward. To further improve the performance of HEM in terms of the solving time, we can set the reward as the negative solving time instead of the negative primal-dual gap integral.

## G.7 TRAINING CURVES

In this section, we provide the training curves of HEM on all nine datasets. The results in Figure 8 show that the performance of our learned policies in terms of the solving time or the primal-dual gap integral drops with the training epochs, demonstrating the effectiveness of our learning process.

## G.8 GENERALIZE TO MORE SETTINGS

### G.8.1 GENERALIZE TO NON-ROOT NODES

Our learned models outperform the baselines for all nodes (both root and non-root nodes) under the one round setting, as shown in Table 17. Specifically, under the one round setting with non-root cuts, our model improves the Time and Primal-dual gap integral by up to 91.29% and 29.61%, respectively.

## G.9 COMPARISON WITH MORE LEARNING-BASED METHODS

We compare HEM with AdaptiveCutsel Turner et al. (2022) and Lookahead Paulus et al. (2022) in Table 15. The experiments demonstrate that HEM significantly outperforms the two learning-based methods by a large margin in terms of the Time (up to 11.21% improvement) and Primal-dual gap integral (up to 24.36% improvement).

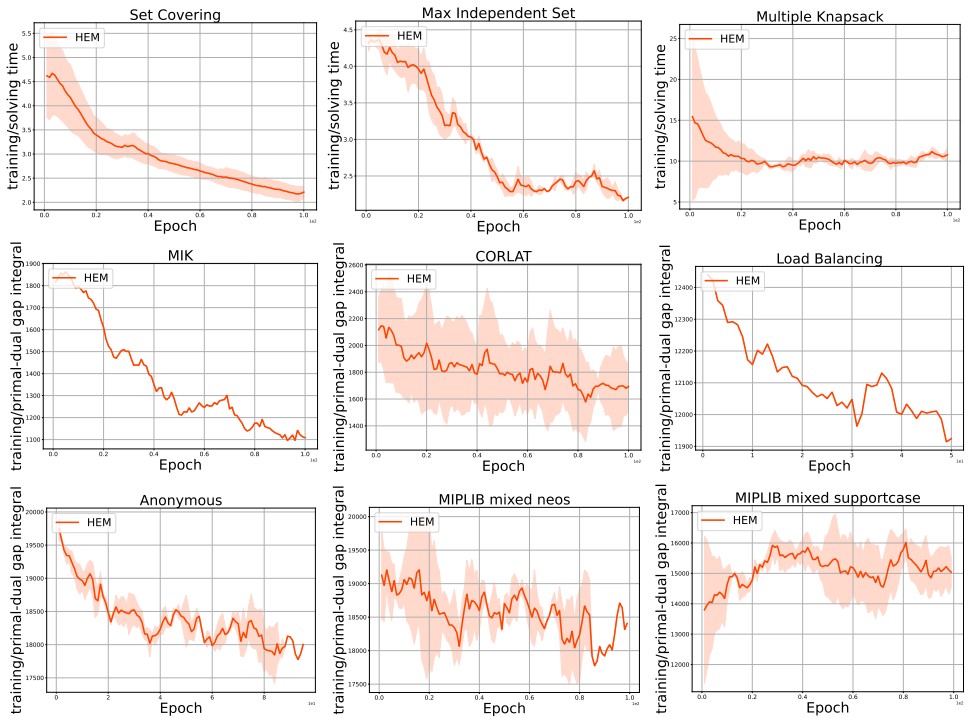

Figure 8: Training curves of HEM on all nine datasets. The x-axis corresponds to the training epochs. The y-axis corresponds to the average solving time on easy datasets and the primal-dual gap integral on the other datasets. The solid curves correspond to the mean and the shaded region to the standard deviation over three random seeds. For visual clarity, we smooth curves.

Table 15: The average performance of HEM and the baselines with AdaptiveCutsel Turner et al. (2022) and Lookahead Paulus et al. (2022). The best performance is marked in bold.

| Method | Maximum Independent Set | | | MIPLIBS mixed setcovering | | |
| | Time(s) | Improvement (Time, %) | PD integral | Time(s) | PD integral | Improvement (PD integral, %) |
|---|---|---|---|---|---|---|
| NoCuts | 8.78 | NA | 71.31 | 170.00 | 9927.96 | NA |
| Default | 3.88 | 55.80 | 29.44 | 164.61 | 9672.34 | 2.57 |
| Random | 6.50 | 26.00 | 52.46 | 165.88 | 10034.70 | -1.07 |
| NV | 7.84 | 10.70 | 61.60 | 161.67 | 8967.00 | 9.68 |
| Eff | 7.80 | 11.10 | 61.04 | 167.35 | 9941.55 | -0.14 |
| SBP | 2.43 | 72.30 | 21.99 | 165.61 | 7408.65 | 25.37 |
| AdaptiveCutsel | 2.74 | 68.79 | 21.40 | 161.03 | 8769.63 | 11.67 |
| Lookahead | 2.27 | 74.15 | 20.89 | **159.61** | 9293.82 | 6.39 |
| HEM (Ours) | **1.76** | **80.00** | **16.01** | 162.96 | **6874.80** | **30.75** |

### G.10 COMPARISON WITH THE SCORE-BASED POLICY (SBP) WITH MORE POWERFUL MODELS

We have conducted the ablation study to show that the performance improvement achieved by HEM is from our novel problem formulation rather than using more powerful models. The results in Table 16 HEM still outperforms the score-based policy (SBP) with more powerful LSTM models, in terms of the Time (up to 80%-67.31%=12.69% improvement) and Primal-dual gap integral (up to 30.75%-22.17%=8.58% improvement).

**Setups.** We implement another baseline, namely SBP with LSTM (SBP+LSTM), which parametrizes the scoring function via an LSTM encoder and a multi-layer perceptron. In terms of model parameters, the model used by SBP with LSTM (172289) is comparable to that of HEM (212749).

Table 16: The average performance of HEM and SBP with LSTM on Maximum Independent Set and MIPLIB mixed supportcase. The best performance is marked in bold.

| Method | Maximum Independent Set | | | MIPLIB mixed supportcase | | |
| | Time(s) | Improvement (Time, %) | PD integral | Time(s) | PD integral | Improvement (PD integral, %) |
|---|---|---|---|---|---|---|
| NoCuts | 8.78 | NA | 71.31 | 170.00 | 9927.96 | NA |
| SBP | 2.43 | 72.30 | 21.99 | 165.61 | 7408.65 | 25.37 |
| SBP+LSTM | 2.87 | 67.31 | 26.01 | 166.25 | 7726.54 | 22.17 |
| HEM(Ours) | **1.76** | **80.00** | **16.01** | **162.96** | **6874.80** | **30.75** |

Table 17: The average performance of HEM and the baselines for non-root nodes under the one round setting. We obtain the results by deploying the models—which are learned at root nodes—to all nodes (root and non-root nodes). The best performance is marked in bold.

| One round with non-root nodes | Easy: Maximum Independent Set | | | Hard: MIPLIB mixed supportcase | | |
| Method | Time (s) | Improvement (Time, %) | PD integral | Time (s) | PD integral | Improvement (PD integral, %) |
|---|---|---|---|---|---|---|
| NoCuts | 8.78 | NA | 71.32 | 170.00 | 9927.96 | NA |
| Default | 1.47 | 83.25 | 13.95 | 128.89 | 9406.43 | 5.25 |
| Random | 2.07 | 76.47 | 20.78 | 124.65 | 9116.01 | 8.18 |
| NV | 3.21 | 63.42 | 29.81 | 134.58 | 8034.05 | 19.08 |
| Eff | 2.08 | 76.27 | 21.06 | **124.20** | 9035.90 | 8.99 |
| HEM (Ours) | **0.76** | **91.29** | **9.75** | 135.80 | **6987.87** | **29.61** |

**Results.** HEM significantly outperforms SBP with LSTM as shown in Table 16. The results demonstrate that HEM significantly outperforms SBP with more powerful models, suggesting that the better performance of HEM is from our novel problem formulation.

## G.11 EXPERIMENTS WITH ADVANCED MODELS

We conduct the following experiments to demonstrate that our method is applicable to advanced models. By replacing the pointer network with the Advanced Model in Kool et al. (2018) (HEM+AM), experiments show that HEM+AM outperforms the baselines (up to 79.61% improvement) as shown in Table 18.

## G.12 GENERALIZE TO OTHER SOLVERS

Our proposed methodology can well generalize to other solvers as shown in Table 19. The results demonstrate that HEM significantly outperforms the default cut selection method in the CBC solver (Saltzman, 2002) in terms of the primal-dual gap (up to 18.67% improvement).

We do not use commercial solvers, such as Gurobi (Bixby, 2007) and (Bliek1ú et al., 2014), as the backend solver, since they do not provide interfaces for users to customize cut selection methods.

As the CBC cannot generate any cut on the dataset Maximum Independent Set, we conduct the experiments on the dataset Load balancing.

We use the primal-dual gap metric rather than the primal-dual gap integral due to the reasons as follows. (1) The primal-dual gap is a well-recognized metric for evaluating the solvers as well. (2) Unlike the SCIP, the CBC does not provide interfaces for users to acquire the primal-dual gap integral. Due to limited time, we do not implement the interface.

## G.13 A DETAILED COMPUTATIONAL ANALYSIS OF OUR MODEL AND THE BASELINES' MODEL

We provide a detailed computational analysis of our proposed model and the baselines' model in Table 20. We summarize the conclusions in the following. (1) The training time of HEM and SBP is comparable, as most of their training time is spent on interacting with solvers to collect training samples. (2) The model parameters of HEM (212749) and SBP with LSTM (172289) are comparable. (3) The inference time of HEM is longer than that of SBP and SBP with LSTM. Nevertheless, the inference time of HEM (0.34s on average) is very low compared to the solving

Table 18: We evaluate HEM with advanced models.

| | Maximum Independent Set | | | MIPLIB mixed supportcase | | |
|---|---|---|---|---|---|---|
| Method | Time (s) | Improvement (%, Time) | PD integral | Time (s) | PD integral | Improvement (%, PD integral) |
| NoCuts | 8.78 | NA | 71.32 | 170.00 | 9927.96 | NA |
| Default | 3.89 | 55.69 | 29.44 | 164.61 | 9672.34 | 2.57 |
| HEM (Ours) | **1.76** | **80.00** | **16.01** | 162.97 | **6874.81** | **30.75** |
| HEM+AM (Ours) | 1.79 | 79.61 | 16.09 | **154.66** | 8440.19 | 14.99 |

Table 19: The performance of HEM and the default strategy used in the CBC solver Saltzman (2002).

| | MIPLIB mixed supportcase | | Load balancing | |
|---|---|---|---|---|
| Method | Primal dual gap (PD gap) | Improvement (%,PD gap) | Primal dual gap (PD gap) | Improvement (%,PD gap) |
| CBC Default | 227.93 | NA | 0.98 | NA |
| HEM (ours) | **185.38** | **18.67** | **0.91** | **7.14** |

time (162s on average), especially on hard datasets. (4) The training of HEM is stable (please see Figure 8).

## G.14 EXPERIMENTS WITH SOME SPECIFIC STRUCTURED MODELS

We have analyzed the selected cuts on Multiple Knapsack (specific structured problems) to show that our learned policies can capture the underlying structure of the specific structured problems. The results in Figure 9 show that our model mainly selects three kinds of cover inequalities, i.e., lifted knapsack cover inequalities (47%) (Gu et al., 1998), lifted minimal cover inequalities (43%) (Gu et al., 1998), and flow cover inequalities (2%) (Gu et al., 1999), for solving Multiple Knapsack problems. Specifically, we analyze the type of cuts selected by our proposed HEM on Multiple Knapsack, a class of problems with specific structures. It is known that a prominent class of cut for knapsack problems is cover inequalities (Gu et al., 1998; 1999). The results demonstrate that our learned policies can select cover inequalities for solving the knapsack problems, suggesting that our model can well capture the underlying structure of specific problems.

## G.15 MEASURING THE PRIMAL AND DUAL INTEGRALS

We have conducted experiments to measure the Primal Integral (PI) and Dual Integral (DI) as shown in Table 21. The results show that the performance improvement of HEM is from both the primal and dual sides.

Specifically, we use the optimal objective values as the reference values to measure the PI/DI. However, it is time-consuming to obtain optimal solutions for all instances. We conduct the experiments on three easy datasets due to limited time. Interestingly, the results demonstrate that proper cut selection policies can improve both the PI and DI. Moreover, the results show that HEM achieves more improvement from the primal side than the dual side on Set Cover and Maximum Independent Set, while HEM achieves more improvement from the dual side on Multiple Knapsack.

Table 20: A detailed computational analysis of our model and the baselines' model.

| | | | Maximum Independent Set | | | | |
|---|---|---|---|---|---|---|---|
| | Model characteristics | | Training | | | Testing | |
| Model | Model paramters | GPU Memory (MB) | Training time (h) | Training samples | Avg Cuts | Inference time (s) | Performance/Time (s) |
| SBP | 18433 | 2.07 | 3.03 | | | 0.0003 | 2.43 |
| SBP+LSTM encoder | 172289 | 2.66 | 2.83 | 3200 | 57 | 0.031 | 2.87 |
| HEM (Ours) | 212749 | 2.81 | 2.54 | | | 0.11 | 1.76 |
| | | | MIPLIB mixed supportcase | | | | |
| | Model characteristics | | Training | | | Testing | |
| Model | Model paramters | GPU Memory | Training time (h) | Training samples | Avg Cuts | Inference time | Performance/PD integral |
| SBP | 18433 | 2.07 | 13.81 | | | 0.0004 | 7408.65 |
| SBP+LSTM encoder | 172289 | 2.66 | 13.24 | 3200 | 173 | 0.033 | 7726.54 |
| HEM (Ours) | 212749 | 2.81 | 13.89 | | | 0.34 | 6874.8 |

Table 21: Evaluate the performance of HEM and Default in terms of the primal and dual integrals.

| | Set Cover | | | |
|---|---|---|---|---|
| Method | PrimalIntegral (PI) | Improvement (%, PI) | DualIntegral (DI) | Improvement (%, DI) |
| NoCuts | 52.34 | NA | 59.85 | NA |
| Default | 45.02 | 13.99 | 49.95 | 16.54 |
| HEM (ours) | **28.84** | **44.90** | **35.95** | **39.93** |
| | Maximum Independent Set | | | |
| Method | PrimalIntegral (PI) | Improvement (%, PI) | DualIntegral (DI) | Improvement (%, DI) |
| NoCuts | 66.83 | NA | 16.24 | NA |
| Default | 32.19 | 51.83 | 11.92 | 26.60 |
| HEM (ours) | **18.33** | **72.57** | **7.97** | **50.92** |
| | Multiple Knapsack | | | |
| Method | PrimalIntegral (PI) | Improvement (%, PI) | DualIntegral (DI) | Improvement (%, DI) |
| NoCuts | 39.39 | NA | 41.62 | NA |
| Default | 25.40 | 35.52 | 25.70 | 38.25 |
| HEM (ours) | **19.24** | **51.16** | **18.90** | **54.59** |

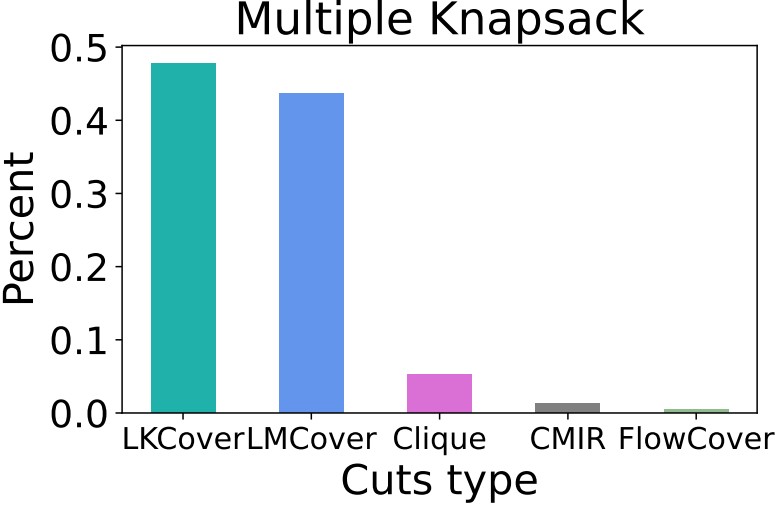

Figure 9: The analysis of selected cuts by our learned model on Multiple Knapsack.

