# OpenReview forum: "Learning Cut Selection for Mixed-Integer Linear Programming via Hierarchical Sequence Model"
_ICLR.cc/2023/Conference — ICLR 2023 poster_

### Official Review · Reviewer_KYdZ · 2022-10-24

**Confidence:** 3
**Correctness:** 3
**Technical Novelty And Significance:** 3
**Empirical Novelty And Significance:** 3
**Recommendation:** 6

**Clarity, Quality, Novelty And Reproducibility:**

**Clarity:** This paper is generally well-organized and easy to follow.

**Quality:** Some crucial justification of the problem formulation and comparison with other learning-based methods are missing.

**Novelty:** While the idea of learning the cut order is novel, the proposed two-level policy model simply incorporates existing models into MILP. The model choice is not clearly discussed.

**Reproducibility:** There is a concern about whether the proposed model can achieve a robust advantage over other learning-based methods.

**Strength And Weaknesses:**

**Strengths:**

+ This paper is generally well-organized and easy to follow.

+ MILP is important for many real-world applications, and learning-based methods have the potential to significantly improve the efficiency of the MILP solver, especially for specific problems from a given distribution. This work is a timely contribution to an important research topic.

+ The idea to predict the cut order for MILP solver is novel.

**Weaknesses:**

**1. Problem Formulation**

Although the idea of cut order prediction is novel, the problem formulation and some design choices are not well supported. To my understanding, the cutting plane operator is important for each branching step in the brand-and-cut algorithm. In other words, cuts can be applied to each node for the search tree. However, this work proposes to only add the learning-based cuts selection at the root node, and run the cut separation with only one single run. As correctly pointed out in this paper, now the problem is formulated as a one-step contextual bandit problem. These simplifications are not well discussed and justified in the paper. Will they significantly hurt the potential and performance of the complete algorithm with a learning-based cut selection at each node with multiple rounds?

If we want to use the learning-based cut selection for each node (not just the root), will a single policy model still perform well for all nodes with a good generalization performance?

**2. Unclear Contribution over Other Learning-based Methods**

In this paper, the proposed method is compared with a few heuristic baselines and one single learning-based method. For the only learning-based method, this work implements a variant version for batched cut selection (rather than the original cut selection in [1,2]). However, since the method proposed in [2] already support batched cut selection, it is unclear why a variant version is needed.

In the implementation, the key score prediction model is a very simple 2-layer MLP trained with evolutionary strategies, which could be less powerful than the model in the proposed method. It is hard to tell whether the better performance of the proposed method is truly from the novel problem formulation or just from using more powerful models.

More importantly, as discussed in this paper, there are many other different learning-based approaches for solving MILP, such as the closely related cut selection methods [3,4]. According to [3], their proposed method can significantly outperform the method in [1]. The other learning-based methods for variable selection, node selection, column generation, and heuristics selection are all potential alternatives. Why none of them are actually compared and analyzed in this work?

The lack of thorough comparisons and analyses with other learning-based methods makes it very hard to evaluate this work's actual contribution and advantage. Given many learning-based alternatives for MILP, why should the user choose the method proposed in this work?

**3. The Policy Model**

It seems that the proposed two-level policy simply incorporates existing models as its policy model. The higher-level model is a simple tanh-Gaussian model, and the lower-level model is the widely-used pointer network where the embedding feature is also already proposed in previous work. No specific model structure or improvement has been proposed for MILP. Is there any structure of MILP that can be further leveraged to build a more efficient model?

The seq2seq pointer network has been significantly improved over the past few years. For example, the input of the pointer network is indeed not an order sequence (which is also the case for cuts in MILP), so the RNN structure is not essential for the encoder. Many advanced model structures like the Attention Model [5], or efficient training methods like POMO [6] have been widely used in the neural combinatorial optimization community. Why does this work choose to use the original less powerful pointer network in the proposed method?

**Other Comments**

1. The proposed method is learned with SCIP in this work. Can the learned policy be generalized well to work with other solvers like Gurobi?

2. The citations of Bengio et al., 2021 for NP-Hardness (page 1), and brand-and-cut (page 3) are not suitable. Please find more suitable references for these citations.

3. It is not suitable to directly use the illustration figure from the original pointer network paper, even in the Appendix (page 22).

[1] Reinforcement learning for integer programming: Learning to cut. ICML 2020.

[2] Learning to select cuts for efficient mixed-integer programming. Pattern Recognition 2021.

[3] Learning to cut by looking ahead: Cutting plane selection via imitation learning. ICML 2022.

[4] Adaptive cut selection in mixed-integer linear programming. Arxiv 2202.10962.

[5] Attention, Learn to Solve Routing Problems! ICLR 2019.

[6] POMO: Policy Optimization with Multiple Optima for Reinforcement Learning. NeurIPS 2020.


**Summary Of The Paper:**

This work proposes HEM, a hierarchical sequence model, to learn the cut selection policies for mixed-integer linear programming (MILP). According to this paper, there are three main issues for cut selection, namely, 1) which cuts should be preferred, 2) how many cuts should be selected, and 3) what order of selected cuts should be preferred. The modern heuristics MILP solvers focus on 1) and 2), while the learning-based solvers are mainly for tackling 1).

This work builds a two-level policy to handle all issues 1) 2) 3) at the same time, which includes a higher-level model to predict the number of cuts, and a lower-level model to select the order of cuts. The higher-level model is a simple tanh-Gaussian model, and the lower-level model is the widely-used pointer network for sequence selection. Experimental results show that the proposed method can outperform several heuristic MILP baselines and one learning-based method.

**Summary Of The Review:**

This work proposes a novel two-level policy model to learn the order of cut selection for MILP. However, due to the current major concerns on problem formulation, unclear contribution over other learning-based methods, and the policy model structure, I cannot vote to accept the current manuscript.

---

> ### Comment · Reviewer_KYdZ · 2022-11-21
> **Thank You for the Thorough Response**
>
> Thank you for the thorough response and additional experiments. I have increased my score to 6.

---

### Official Review · Reviewer_xxrL · 2022-10-24

**Confidence:** 3
**Correctness:** 3
**Technical Novelty And Significance:** 3
**Empirical Novelty And Significance:** 3
**Recommendation:** 8

**Clarity, Quality, Novelty And Reproducibility:**

Other comments:
* Figure 1: What is the time limit?
* Can the authors provide some more justification as to why they focus solely on a single round of separation at the root, as opposed to separation throughout the tree? Do the authors believe the methods or conclusions of the paper would differ in interesting ways in this (more realistic) setting?
* p8: What is the "original index" of the generated cuts?

**Strength And Weaknesses:**

The paper is well-written and is a worthwhile contribution to an interesting and relevant stream of research on cut selection and MILP solver configuration more broadly.

While the authors choice of primal-dual integral as the metric of choice for the method comparisons, I think the paper would benefit from some additional quantitative measurements in the computational study. This additional data (and accompanying analysis) could help give more insight into _how_ the new cut selection method is improving SCIP. For example, also measuring the primal and dual integrals would potentially give some insight into whether the gains are coming from the primal side (e.g., better aligned with SCIP's heuristics) or the dual side (e.g., better aligned with SCIP's tree search algorithm). Additionally breaking out the node count or simplex iteration count would provide insight into how much work SCIP is expending on the node LP solves.

I do not think this further analysis is necessary for all of the computational results, but it would be welcome in certain spots. In particular the ablation study would benefit from more of a breakdown to understand what each component of the proposed method is affecting the underlying solver.

Beyond this, my biggest lingering questions relate to how these results will generalize to other solvers, and what exactly "cut ordering" means in the implementation.

The authors write: "RandomAll...randomly permutes all the candidate cuts, and adds all the cuts to the LP relaxations in the random order." In your implementation, are you explicitly adding the cuts to the LP relaxation? Or are you instead registering the cuts through the "Separator" API, or filtering them through the "Cut Selector" API? If it is through one of these higher-level SCIP APIs, have the authors verified that all the cuts are added "as-is" to the LP relaxation in the given order, or is SCIP potentially applying some filtering/transformations/reordering to the cuts registered in the pool before adding to the LP? If the latter, is there some order dependence in SCIPs internals (e.g., SCIP keeps a running tally of some type of condition number as it takes a linear pass through the registered cuts and filters based upon it).

The reason I belabor this a bit is that I think it can give some insight into whether the method is learning how to make SCIP work well, as opposed to learning something more general about the instances in a somewhat solver independent fashion (e.g., the method is learning how to order cuts to make the underlying linear algebra more efficient). If the method is doing the latter, this makes the case that the results will plausibly generalize to other solvers (the current paper does not make a strong argument to this effect, understandably so given the sheer implementation complexity such a study would entail).

**Summary Of The Paper:**

This paper proposes an algorithm for cut selection in a mixed-integer linear programming (MILP) solver. The algorithm aims to determine which subset of a collection of proposed cuts to add at the root node, and in what order they should be added. To accomplish this, the authors propose a hierarchical reinforcement learning-based method that first determines the number of cuts to add, and then determines  which cuts to add and in what order.

**Summary Of The Review:**

The authors successfully build on prior work on cut selection for MILP solvers. While I have a few points where I think the paper could be further clarified or improved with greater insight, I generally think it is a nice contribution to the literature.

---

### Official Review · Reviewer_j8AA · 2022-10-24

**Confidence:** 4
**Correctness:** 4
**Technical Novelty And Significance:** 2
**Empirical Novelty And Significance:** 2
**Recommendation:** 6

**Clarity, Quality, Novelty And Reproducibility:**

The paper is clearly written and to the best of my understanding, the experiments are reproducible conditioned on the release of code and data. The technical quality of the paper is reasonable, however, the improvements are rather incremental. Given the numerous successes of reinforcement learning on tasks of similar nature, it's a reasonably safe bet that something like this would work better than what is out there. What would make this a more interesting paper is some type of argument that the state representation captures something important about the solution structure. What one would imagine as an intelligent constraint generation method is something that represents the abstract high-level structure of the solution (e.g., this solution is incomplete/non-integral because it's missing some sort of "feature") and is able to generate entire structures of constraint that resolve these missing features. The authors could consider experimenting with some specific structured models and show whether the generated constraint lists can be mapped back to the high-level structure of the problem and/or solution.

**Strength And Weaknesses:**

Strengths:
* This paper shows progress on an important problem.
* The idea is sensible, making use of the observation that order matters for constraints and proposing a solution that addresses this.

Weaknesses:
* While the paper does a good job of comparing the performance of the proposed method to prior work and ablations, there are some really important questions that seemed to be left open. What is the computational cost of the training? Is it stable? How large are the trained models? How long does inference take? It struck me that the main competitor in comparisons (the scoring based policy) is based on a rather small model (which is an important detail and buried all the way on page 20). I then tried to find whether the HEM model is comparable in terms of parameters and training time, and could not find these details. This raises questions as to whether the observed gains are due to a more efficient problem representation, or due to model disparity. This analysis is important and needs to be prominently displayed.


**Summary Of The Paper:**

This paper proposes a method for constraint generation within the branch-and-cut framework. Differing from prior work, the paper casts the constraint generation problem as an MDP and proposes a hierarchical policy to generate a tuple of cuts of variable length depending on problem state. The constraint generation policy is a stochastic bi-level policy which first selects the number of cuts to generate, then selects the list of cuts with desired cardinality using a pointer network. The state of the problem is represented by the model generated so far and the set of candidate constraints and is encoded via an LSTM due to it's variable-length nature. The policy is trained using policy gradient.  The results are evaluated on standard benchmarks.

**Summary Of The Review:**

This is a technically reasonable paper where novelty is somewhat limited, and at present marginally above threshold. I think this novelty/insight can be improved substantially by trying to interpret what these policies do and argue that that they implement some type of abstract reasoning.

---

### Decision · Program_Chairs · 2023-01-20

**Decision:**

Accept: poster

**Justification For Why Not Higher Score:**

Technical novelty is not high with respect to the general literature on ML for combinatorial optimization and ML for solving MILP problems within the branch-and-cut framework.

**Justification For Why Not Lower Score:**

The paper uses the key insight that ordering of constraints matter and uses an appropriate policy representation to show improvements on an important problem.

**Metareview: Summary, Strengths And Weaknesses:**

This paper considers the problem of cut selection within the branch-and-cut framework to solve mixed-integer linear programming (MILP) instances. It learns a bi-level constraint generation policy: 1) Select the number of cuts to generate, and 2) select the list of cuts with desired cardinality using a pointer network. The policy is trained within the RL framework using a policy gradient method. Experimental results show good improvements.

All the reviewers' agree that the paper looks an important problem and shows good progress by relying on the key insight that ordering of constraints is critical even though technical novelty is not high. Overall, the paper is very well-written and easy to follow. The reviewers' provided some constructive suggestions and asked clarifying questions. Authors' have done a very good job in answering the questions during the rebuttal phase.

I recommend accepting the paper and strongly encourage the authors' to revise the paper to reflect the reviewer-author discussion for the final submission.

**Note From Pc:**

if the above contains the word "oral" or "spotlight" please see: "oral" presentation means -> notable-top-5% and "spotlight" means -> notable-top-25%. As stated in our emails, we are disassociating presentation type from AC recommendations